



# Towards spatio-temporal comparison of transient simulations and temperature reconstructions for the last deglaciation

Nils Weitzel[1], Heather Andres[2], Jean-Philippe Baudouin[1], Marie Kapsch[3], Uwe Mikolajewicz[3], Lukas Jonkers[4], Oliver Bothe[5], Elisa Ziegler[1,6], Thomas Kleinen[3], André Paul[4], and Kira Rehfeld[1,6]

[1]Department of Geosciences, University of Tübingen, Tübingen, Germany
[2]Northwest Atlantic Fisheries Centre, Fisheries and Oceans Canada, St. John's, Newfoundland, Canada
[3]Max Planck Institute for Meteorology, Hamburg, Germany
[4]MARUM Center for Marine Environmental Sciences, University of Bremen, Bremen, Germany
[5]Formerly at Helmholtz-Zentrum Hereon, Institute of Coastal Systems - Analysis and Modelling, Geesthacht, Germany
[6]Department of Physics, University of Tübingen, Tübingen, Germany

**Correspondence:** Nils Weitzel (nils.weitzel@uni-tuebingen.de)

**Abstract.** An increasing number of climate model simulations is becoming available for the transition from the Last Glacial Maximum to the Holocene. Assessing the simulations' reliability requires benchmarking against environmental proxy records. To date, no established method exists to compare these two data sources in space and time over a period with changing background conditions. Here, we develop a new algorithm to rank simulations according to their deviation from reconstructed

magnitudes and temporal patterns of orbital- as well as millennial-scale temperature variations. The use of proxy forward modeling avoids the need to reconstruct gridded or regional mean temperatures from sparse and uncertain proxy data.

First, we test the reliability and robustness of our algorithm in idealized experiments with prescribed deglacial temperature histories. We quantify the influence of limited temporal resolution, chronological uncertainties, and non-climatic processes by constructing noisy pseudo-proxies. While model-data comparison results become less reliable with increasing uncertainties,

we find that the algorithm discriminates well between simulations under realistic non-climatic noise levels. To obtain reliable and robust rankings, we advise spatial averaging of the results for individual proxy records.

Second, we demonstrate our method by quantifying the deviations between an ensemble of transient deglacial simulations and a global compilation of sea surface temperature reconstructions. The ranking of the simulations differs substantially between the considered regions and timescales. We attribute this diversity in the rankings to more regionally confined temperature

variations in reconstructions than in simulations, which could be the result of uncertainties in boundary conditions, shortcomings in models, or regionally varying characteristics of reconstructions such as recording seasons and depths. Future work towards disentangling these potential reasons can leverage the flexible design of our algorithm and its demonstrated ability to identify varying levels of model-data agreement.

## 1 Introduction

Major boundary condition changes make the transition from the Last Glacial Maximum ($\sim$ 21 ka, LGM) to the current warm period, the Holocene interglacial (starting at $\sim$ 11.65 ka), an important period for understanding past global warming episodes





and a valuable period for testing climate models. This transition, called the last deglaciation (LD), is the most recent period with natural radiative forcing variations of comparable magnitude to projected anthropogenic emissions. During the LD, the configuration of orbital parameters changed, resulting in a minimum in Northern Hemisphere summer insolation around 24 ka and a maximum around 11 ka (Berger, 1978). The $CO_2$ concentration increased from $\sim 185$ ppm to $\sim 280$ ppm (Köhler et al., 2017), and sea level rose by $\sim 100$ m (Lambeck et al., 2014) because large ice sheets over North America (the Laurentide and Cordilleran ice sheets) and Europe (the Fennoscandian and British ice sheets) retreated entirely (Batchelor et al., 2019).

In recent years, the LD has been simulated with an increasing number of climate models that apply transiently changing boundary conditions (Ivanovic et al., 2016). Proxy-based temperature reconstructions suggest that (near-)surface temperatures increased at most places during the LD (Cleator et al., 2020; Paul et al., 2021) and by 3-8 K in the global mean (Annan et al., 2022; Tierney et al., 2020). Most climate models simulate LGM global mean surface air temperature (GMSAT) anomalies in this range (Kageyama et al., 2021). However, proxy evidence suggests that considerable regional differences exist in the magnitude and temporal pattern of the deglacial temperature changes (Clark et al., 2012). So far, it has not been quantitatively assessed whether climate models can not only reproduce the reconstructed GMSAT changes, but also the spatial fingerprint of the temperature evolution when forced with appropriate boundary conditions. This assessment is challenging because it relies on sparse and indirect observations of past climate and uncertain boundary conditions (Ivanovic et al., 2016).

Previous model-data comparison efforts involving global databases of proxy records focused on time slices (e.g. Hargreaves et al., 2013; Harrison et al., 2014) or the Common Era (e.g. PAGES 2k-PMIP3 group, 2015; PAGES 2k Consortium, 2019). They quantify either differences between two distinct states (e.g. LGM vs. pre-industrial) or fluctuations during a stationary climate state (e.g. magnitude of temperature variability). So far, transient simulations of the LD have only been compared against a small number of selected proxy records or large-scale mean reconstructions (e.g. Liu et al., 2009; Menviel et al., 2011; He et al., 2021; Dallmeyer et al., 2022). Here, we develop a model-data comparison algorithm that compares LD simulations with temperature reconstructions in space and time. In particular, our algorithm allows to quantitatively assess the following four questions:

1. Is the magnitude of simulated deglacial warming in agreement with reconstructions?

2. Is the temporal pattern of the glacial-to-interglacial (called orbital-scale) warming trend accurately simulated?

3. Are the magnitudes of simulated millennial-scale variations modulating the warming trend similar to reconstructions?

4. How much does the temporal pattern of simulated millennial-scale variations deviate from reconstructions?

We analyze the four components of the deglacial temperature evolution associated with these questions separately because the robustness of their reconstruction varies, and they are potentially controlled by different mechanisms and uncertain boundary conditions. In the following, we call these four components the '*orbital magnitude*' (magnitude of orbital-scale temperature variations), '*orbital pattern*' (temporal pattern of orbital-scale variations), '*millennial magnitude*' (magnitude of millennial-scale variations), and '*millennial pattern*' (temporal pattern of millennial-scale variations). Note that throughout this paper we use the term 'orbital' to describe climate variations occurring on similar timescales ($\sim 6$ kyr and longer) to variations



in the Earth's orbital configuration, although changes in greenhouse gas (GHG) concentrations and ice sheets are the main contributors to radiative forcing on these timescales during the LD.

To illustrate our model-data comparison algorithm, we use a global database of sea surface temperature (SST) reconstructions and an ensemble of LD simulations (Sect. 2). SSTs are reconstructed from geochemical indices and species assemblages extracted from marine sediment cores. Both reflect the climate state at the time of deposition (Jonkers et al., 2020). However,

the reconstructed temperatures are also influenced by non-climatic processes during the recording of the temperature signal, the archival of the sensors in the sediment, and the measurement of the sensors. These include imperfect calibrations to temperature, biases from confounding environmental variables, deviations from mean annual SST through seasonal and habitat depth preferences, temporal smoothing by bioturbation, noise from using a small number of short-living replicates, measurement errors, and chronological uncertainties (MARGO Project Members, 2009; Jonkers and Kučera, 2017; Dolman and Laepple,

2018; Jonkers and Kučera, 2019; Osman et al., 2021).

These non-climatic processes create a challenge for model-data comparison: whether a simulation produces a more realistic climate evolution than others is not necessarily the same as finding the simulation that minimizes the difference to a set of reconstructions, since reconstructions are an imperfect representation of the actual climate evolution. To obtain a representation of the simulated climate that is comparably disturbed by non-climatic processes as reconstructed SSTs, we use proxy system

models (PSMs). PSMs are mathematical descriptions of the processes involved in the recording, archiving, and measurement of the response of an environmental proxy to the climate (Evans et al., 2013). PSMs are applied to climate simulation output to create forward-modeled proxy time series, which mimic the properties of real proxies. A comparison of these forward-modeled proxy time series against proxy-based reconstructions facilitates a more consistent comparison under the assumption that real and modeled proxies are subject to comparable modifications (Laepple and Huybers, 2014; Dee et al., 2017; Bühler et al.,

75 2021).

A second challenge in model-data comparison is to separate mismatches between simulations and reconstructions due to uncertain boundary and initial conditions, poorly constrained model parameters, and imperfect or missing representations of relevant processes by climate models (Braconnot et al., 2012). This challenge could in principle be assessed through large model ensembles, but computational resources are insufficient to produce them. Therefore, we focus here on incorporating

methods to account for uncertainties from imperfect reconstructions.

The goals of this paper are threefold. First, we motivate and present our proposed model-data comparison algorithm (Sect. 3.1). Second, we test our algorithm with pseudo-proxy experiments (PPEs; von Storch et al., 2004), in which the deglacial climate evolution is prescribed by a reference simulation (Sect. 3.3, 4.1). These experiments help us to understand the characteristics of our algorithm and to assess its reliability and robustness under limited temporal resolutions, chronological uncertainties,

and non-climatic modulations of the proxy records in idealized experiments. To our knowledge, model-data comparison algorithms have never been systematically tested with PPEs. Third, we demonstrate our method by quantifying the deviations between forward-modeled proxy time series derived from ten LD simulations and the global compilation of SST reconstructions (Sect. 4.2). Finally, we discuss implications and limitations of our results, and outline future work (Sect. 5).



## 2  Data

### 2.1  Transient simulations

We use ten simulations from three climate models which all simulate the period 22 ka to 6 ka (Fig. 1, Table 1). Six simulations were run with MPI-ESM-CR (Kapsch et al., 2022; Kleinen et al., 2022, 2023). In these simulations, GHG concentrations and orbital parameters were updated transiently. Ice sheet topographies are changed according to the GLAC-1D or ICE-6G reconstructions (see Table 1). Meltwater from ice sheets is either transported into the ocean using dynamic river routing (Riddick et al., 2018), distributed uniformly over all grid cells, or removed from the system (see Table 1). MPI_Glac1D_PTK uses a parameter configuration that leads to a smaller LGM to Holocene global-mean near-surface air temperature (GMSAT) difference than the other MPI-ESM simulations. Furthermore, atmospheric parameters in the 'P3' simulations are slightly different from those in the 'P2' simulations to correct a pre-industrial cold bias (Kapsch et al., 2022). We further include three CCSM3 simulations from the TraCE-21ka project (Liu et al., 2009). In TraCE-ALL, orbital parameters, GHG concentrations, ICE-5G ice sheet topographies, and manually prescribed meltwater fluxes are adapted transiently. In TraCE-GHG, all boundary conditions except for GHG concentrations are fixed at the 22 ka state of TraCE-ALL. Similarly, only orbital parameters are changed in TraCE-ORB. Finally, we use the ALL-5G simulation from the QUEST FAMOUS last glacial cycle ensemble (Smith and Gregory, 2012). Orbital parameters, GHG concentrations, and Northern Hemisphere ICE-5G ice sheet topographies are updated transiently. In contrast to the other simulations, the Antarctic ice sheet topography and land-sea mask are fixed to pre-industrial values. No meltwater fluxes are applied. The boundary conditions were applied with an acceleration factor of 10. More information on the simulations are provided in the Supplement (Text S2).

The simulation ensemble has a large spread in the four components of the deglacial temperature evolution described in Sect. 1 (Fig. 1). In the simulations with changing GHG concentrations, ice sheets, and orbital parameters, the deglacial GMSAT increase is between $\sim 3$ K in FAMOUS and $\sim 6.5$ K in MPI_Glac1D_P3. With $\sim 1$ K in TraCE-ORB and $\sim 3$ K in TraCE-GHG, the deglacial warming is lower in the two single forcing experiments. Deglacial warming starts later in TraCE and FAMOUS than in MPI-ESM, and the warming trend is smoother in MPI-ESM than in TraCE-ALL. Two different aspects of meltwater injection appear to play an important role in the GMSAT histories of these runs: the method of application and the progression through time. Simulations without meltwater fluxes feature weak millennial-scale fluctuations (e.g. MPI_Ice6G_P2_noMWF), and simulations with locally applied meltwater fluxes (e.g. MPI_Ice6G_P2) generate stronger GMSAT fluctuations than the simulation with global injection (MPI_Ice6G_P2_glob). Differing meltwater histories lead to an abrupt warming at $\sim 14.5$ ka in TraCE-ALL but cooling events in the MPI-ESM experiments with meltwater input.

### 2.2  Sea surface temperature reconstructions

We use temperature reconstructions from the PalMod 130k marine paleoclimate data synthesis v1.1.1 (Jonkers et al., 2023), which is a compilation of proxy records derived from marine sediment cores. V1.1.1 is an update from Jonkers et al. (2020) with 252 published (near-)surface temperature time series covering various parts of the last glacial cycle. As described in Jonkers et al. (2020), age models were harmonized using the Bayesian age modeling algorithm BACON (Blaauw and Christen,





2011). For each sediment core, 1000 iterations of the age-depth model were saved in the database to quantify chronological
uncertainties. The database combines temperature reconstructions from multiple sensors which were taken unchanged from
the original publications. For some proxy records, reconstructions from different original publications are included in the

database. We retain all records from the same sediment cores if they are based on different sensors. We average reconstructions
originating from the same sediment core and sensor if all sample depths coincide. If the depths differ, we select the time series
covering the longest period during the deglaciation. Reconstructions from the same proxy data but calibrated for different
seasons are averaged to obtain pseudo-annual temperatures. More details on the preprocessing of the proxy records are provided
in the Supplement (Text S3).

We select all (near-)surface temperature samples in the interval 22-6 ka from the database. Although some of the recon-
structions are representative of subsurface conditions, we denote them as sea surface temperature (SST) reconstructions in the
following. To compute robust statistics, we use only time series with at least 10 samples, which cover more than 8 kyr and have
a mean temporal resolution of at least 1 kyr. 74 temperature records from 50 unique sediment cores satisfy these conditions
(Fig. 1b, Table 2). Most of them are located on continental margins with the biggest clusters located in the North Atlantic

and the Indo-Pacific Warm Pool. 38 temperature records are reconstructed from Mg/Ca, 17 from $U^k_{37}$, 17 from planktonic
foraminifera assemblages, 1 from $TEX_{86}$, and 1 from diatom assemblages. Unlike some recent studies focusing on specific
sensors (e.g. Paul et al., 2021; Osman et al., 2021), we employ a multi-sensor approach using the calibrations proposed by the
original authors. We make this choice because the number of records in the database is too small to focus on a specific sensor
and sensors tend to be regionally clustered which makes a systematic assessment of differences between them unfeasible within

our study design. For more discussion on the differences between sensors see Paul et al. (2021) and the references therein.

## 3   Methods

This section first presents our model-data comparison algorithm (Sect. 3.1). The algorithm employs a simple PSM with two
parameters which we estimate in Sect. 3.2. Sect. 3.3 describes the PPEs for assessing the reliability and robustness of our
algorithm.

### 3.1   Model-data comparison algorithm

Our model-data comparison algorithm consists of four main steps as visualized in Fig. 2. We provide technical descriptions of
the steps in the next four subsections but first motivate them here:

**1) Compute forward-modeled proxy time series from simulation output.**

To compare simulations and reconstructions, we have to bridge the gaps between the two types of data in terms of spatio-

temporal coverage and non-climatic influences on the proxy measurements. This is done in a forward approach, in which a
PSM is applied to simulation output. The PSM output, which we call 'forward-modeled proxy time series', is compared to the
measured proxies. Alternatively, inverse approaches infer gridded temperature fields from reconstructions using interpolation
in space and time (Tingley et al., 2012). We choose the forward approach, because it follows the natural process-chain from



the climate signal to the sample measurements (Evans et al., 2013) and it avoids the estimation of spatio-temporal temperature

correlation structures, which are hard to estimate from sparse proxy data (Tingley et al., 2012). We compare measured and

forward-modeled proxy time series in temperature units instead of measured proxy units, because it allows averaging deviations

from different sensors and no established forward calibrations exist for assemblage-based reconstructions.

**2) Decompose time series into magnitudes and temporal patterns of timescale-dependent variations.**

We decompose each temperature time series into four components, each of which is designed to assess one of the four

questions posed in Sect. 1. We assess the deviations between forward-modeled proxy time series and proxy records for each

component separately because computing a single score for the deviation between simulations and reconstructions is prone to

conceal sources of discrepancies. For example, a simulation could simulate the spatio-temporal temperature pattern accurately

but receive a poor score due to an under-estimation of the LGM-to-Holocene temperature change.

**3) Quantify deviations between reconstructions and forward-modeled proxy time series for individual proxy records.**

Accounting for uncertainties associated with the SST reconstructions and simulations requires a probabilistic comparison

framework. We implement such a framework using a Monte Carlo (MC) approach, which propagates uncertainties through

the algorithm. The deviations between the resulting probability distributions for forward-modeled proxy time series and the

corresponding reconstructed SST records are quantified with a distance function that takes into account the full probability

distributions, including multi-variate distributions such as those corresponding to auto-correlated time series, and not just

summary statistics like the mean or standard deviation. Applying the distance function to the respective probability functions

results in a single number for the deviation between forward-modeled proxy time series and reconstructions for each of the

four components in which we decompose the time series in step 2.

**4) Average deviations in space.**

Deviations between forward-modeled proxy time series and reconstructions can depend strongly on the unknown manifesta-

tion of non-climatic influences in the measured proxies. Assuming that most non-climatic processes are uncorrelated between

proxy records, the influence of these processes can be reduced by spatially averaging deviations computed for individual proxy

records. Computing averages in this last step instead of averaging temperature time series in the beginning avoids interpolating

proxy records with irregular time axes to a common resolution.

When running the algorithm for an ensemble of simulations and a set of proxy records, steps 1 to 3 are performed sequentially

for all combinations of proxy records and simulations. Therefore, we describe these steps for one example proxy record and

simulation below, while step 4 combines multiple records to a spatially averaged score.

### 3.1.1 Step 1: compute forward-modeled proxy time series

We employ a simple PSM that takes simulated 3D (lon × lat × time) mean annual SST fields ($C_{\text{Sim}}$) as input and modifies them

to resemble a reconstructed SST record ($C_{\text{FM}}$, FM = forward-modeled). The PSM consists of three steps, spatial interpolation

($P_{\text{space}}$), temporal downsampling ($P_{\text{time}}$), and an additive noise process ($\varepsilon$):

$$C_{\text{FM}} = P_{\text{time}}\left(P_{\text{space}}(C_{\text{Sim}})\right) + \varepsilon. \tag{1}$$





First, we interpolate the spatial SST fields bilinearly to the proxy record location. Given the smoothness of SST fields on long time scales, the influence of the specific interpolation method is negligible. For the downsampling of the simulated time series to the time axis of the proxy record, we draw $N$ MC samples of simulated and reconstructed time series to quantify chronological uncertainties. For each MC sample, we randomly select one iteration of the age-depth model (see Sect. 2.2) and downsample the simulated time series to the irregular time axis of the proxy record using blocksampling. The blocksampler cuts the simulated time series into disjoint slices with cutting dates at the midpoints between the sample ages, and assigns the averaged signal of each slice to the date of the corresponding sample. This procedure imitates the limited temporal resolution and integrated nature of the proxy records. The result of the temporal downsampling is a temporally aligned set of $N$ reconstructed SST time series (Fig. 2, top left) and $N$ time series of downsampled SST simulations. The blocksampling strategy assumes that gaps in the sampling of records are smaller than the depths over which individual samples average, at least after accounting for smoothing from bioturbation. More detailed reporting of the top and bottom sampling depths of each sample could be used to refine the downsampling procedure and quantify its influence.

In our PSM, we summarize the effects of the inherent uncertainties of SST reconstructions (see Sect. 1) by a Gaussian additive noise process with a specified signal-to-noise ratio (SNR) and temporal autocorrelation structure. For each of the $N$ time series of downsampled SST simulations, we add a random realization of the additive noise process to the SST time series. We call the $N$ resulting time series 'forward-modeled proxy time series' (Fig. 2, top right). We use the additive noise approach because metadata is missing to explicitly model processes that lead to deviations of the reconstructions from mean annual SSTs for all records in the compilation. In addition, climate models do not simulate all variables required to model these processes. For example, the recording season and depth of the sensors are uncertain, insufficiently reported in the literature, and might vary over the LD due to habitat tracking (Mix, 1987; Jonkers and Kučera, 2017). Therefore, we compare all (near-)surface temperature reconstructions with mean annual SSTs from the simulations. As we only analyze SST changes over time, offsets from mean annual SSTs in the absolute reconstructed temperatures, which stay (nearly) constant over time, are not affecting our results.

### 3.1.2 Step 2: decompose time series

In step two, we extract the four components of the deglacial SST evolution outlined above: magnitudes and patterns for both orbital- and millennial-scale variations. We first decompose each of the $N$ time series into three timescales with Gaussian smoothers (Fig. 2, second row; see Supplement Fig. S2-S9 for further examples of timescale decompositions). We use Gaussian smoothers because they are a robust method for the analysis of irregularly spaced time series in the time and frequency domain (Rehfeld et al., 2011). The three timescales are orbital-, millennial-, and sub-millennial-scale variations, whose ranges abut one another. We select a smoothing period of 1 kyr to separate sub-millennial from millennial timescales. Since there is no clear scale separation between millennial and orbital variations, we employ three smoothing periods, 4 kyr, 6 kyr, and 8 kyr, and average the respective quantified deviations after step 4.

Next, we isolate the temporal patterns of the variations from their magnitudes. To this purpose, we compute the standard deviations of all reconstructed and forward-modeled proxy time series, which are a measure of the magnitude of variations on





a given timescale. As we obtain one estimate from each MC sample, this leads to probability distributions for the timescale-dependent magnitudes of variations in reconstructed and forward-modeled proxy time series. We define the pattern of the respective variations as the normalized, i.e. centered and standardized, time series. We obtain $N$ realizations of normalized time series which we interpret as $M$-dimensional probability distributions, where $M$ is the number of samples of the respective

proxy record. Thus, the decompositions result in eight probability distributions, four for the reconstructed and forward-modeled proxy time series respectively (orbital and millennial magnitudes as well as patterns, Fig. 2, third row). Each of the distributions is represented by $N$ MC samples.

### 3.1.3 Step 3: quantify deviations between forward-modeled proxy time series and reconstructed SST records

In the third step, we compute the deviation between the simulated forward-modeled proxy time series and reconstructions for

each proxy record and each of the four components (Fig. 2, bottom row). Each of these deviations is quantified with the integrated quadratic distance (IQD). The IQD is a proper divergence function that has desirable mathematical properties for model selection as it penalizes overly confident or conservative uncertainty estimates compared to the unknown true uncertainties (Thorarinsdottir et al., 2013). The IQD is applicable for univariate and multivariate probability distributions. It is defined as

$$\text{IQD}(\mathbb{P}, \mathbb{Q}) \;=\; \frac{1}{M}\mathbb{E}_{\mathbb{P},\mathbb{Q}}|X - Y| - \frac{1}{2M}\left(\mathbb{E}_{\mathbb{P}}|X - X'| + \mathbb{E}_{\mathbb{Q}}|Y - Y'|\right), \tag{2}$$

where $\mathbb{P}$ is the probability distribution of forward-modeled proxy time series, $\mathbb{Q}$ is the probability distribution of the reconstructions, $M$ is the dimension of $\mathbb{P}$ and $\mathbb{Q}$, and $\mathbb{E}$ denotes expected values. Further, $X$ and $X'$ are independent random variables distributed according to $\mathbb{P}$, and $Y$ and $Y'$ are independent random variables distributed according to $\mathbb{Q}$. The first term in Equ. (2) is the expected difference between draws from the distributions of forward-modeled proxy time series ($\mathbb{P}$) and reconstructions ($\mathbb{Q}$). The two last terms quantify the spread of the distributions $\mathbb{P}$ and $\mathbb{Q}$ since $\mathbb{E}_{\mathbb{P}}|X - X'|$ is the expected difference

between two random draws from the distribution $\mathbb{P}$. The name IQD is motivated by the fact that in one dimension, the IQD is equal to the integral over the squared difference between the cumulative distribution functions of $\mathbb{P}$ and $\mathbb{Q}$.

The IQD takes positive values ($\text{IQD}(\mathbb{P}, \mathbb{Q}) \geq 0$). It is only zero when $\mathbb{P}$ and $\mathbb{Q}$ are equal ($\text{IQD}(\mathbb{P}, \mathbb{P}) = 0$). Smaller IQD values imply a smaller deviation and thus a better agreement of forward-modeled proxy time series and reconstructions. In the absence of age and proxy uncertainties, the IQD reduces to the mean absolute difference between numbers (magnitudes) or time series

(patterns). The IQD can be applied to quantities of arbitrary units. In our case, the units are temperature [K] for the comparison of magnitudes, and standard deviations [$z$] for patterns. We compute the IQD using a MC approximation of Equ. (2) with the MC samples from step 2. Numerical tests determined that IQD estimates are stable for $N \geq 100$ (see Supplement). Therefore, we use $N = 100$ for the computationally demanding PPEs and $N = 1000$ for the real-world application. Computational details are provided in the Supplement (Text S4).

### 3.1.4 Step 4: average deviations in space

We analyze IQDs averaged on four spatial scales: locally, regionally (see color-coding of dots in Fig. 1b for the assignment of proxy records to the regions considered in this study), zonally, and globally. For local IQDs, we treat each proxy record individ-





ually, i.e. without averaging proxy records from the same core or nearby locations. Zonal IQDs are obtained by averaging over proxy records within overlapping bands of $20°$ width that move in $5°$ steps (Fig. 2, bottom row). We only consider latitudinal

bands containing at least five proxy records to only incorporate spatial averages where we can assume that a substantial amount of non-climatic influences is averaged out.

### 3.2    Estimation of proxy system model parameters

The PSM described in Sect. 3.1.1 requires a SNR parameter quantifying the ratio between climatic and non-climatic variations and the specification of a temporal autocorrelation structure of the additive Gaussian noise process. Previous studies only

estimated SNRs and autocorrelations for a subset of our sensors ($U^k_{37}$, Mg/Ca) on sub-orbital timescales (Laepple and Huybers, 2014; Reschke et al., 2019). Therefore, we estimate the PSM parameters using the SST reconstruction database (see Sect. 2.2).

To obtain these estimates, we decompose the SST records into a similar structure as Equ. (1), i.e. the sum of a local mean SST signal $P_{\text{space}}(C)$ and a realization of a Gaussian noise process $\varepsilon$, which aggregates all deviations from the local mean SST signal. The decomposition starts by constructing clusters of SST records centered around each of the 74 SST records

selected from the database. The clusters contain the records within a radius of $l \in \{100, 200, ..., 1000\}$ km around the central record (see Fig. 3 for an example cluster with $n = 3$ records centered around record SO201_2_12KL). For each cluster, we construct a local mean signal by averaging over the records in the cluster (red line in Fig. 3a). More specifically, we interpolate nearby records to a regular temporal resolution of 100 yrs, center the records, and average over the resulting time series. We use the mean age model of each record and not the age ensemble members since we account for chronological uncertainties at

a different step of the PSM. Using the age ensembles instead of the mean ages strongly reduces the estimated SNR and likely biases it low (not shown). Note that we average records of different temporal resolutions which tends to underestimate high frequency contributions to $\varepsilon$. However, all records have at least a millennial resolution such that the relevant millennial and orbital timescales should be less affected by the interpolation and subsequent averaging.

For the record in the center of the cluster, we compute the residual from the local mean signal (green line in Fig. 3b) which

is treated as a realization of the Gaussian noise process ($\varepsilon$ in Equ. 1). We compute the variance ratio between the local mean signal and the residual which provides an estimate of the SNR. Due to the short time series length, the structure of the temporal autocorrelation cannot be determined from the residuals. We choose to describe $\varepsilon$ as an autoregressive process of order one (AR1) because it is determined by only two parameters and as a compromise between a white noise process without temporal autocorrelation and power-law processes with long-range autocorrelations. This AR1 process is specified by the SNR and a

decorrelation length, which we estimate from the residual. We iterate this process for all 74 records if the clusters around the respective records contain at least a specified number of records. Then, we take the medians of the SNRs and the decorrelation lengths in all clusters to reduce the noise in the parameter estimates which results from the predominantly small cluster sizes (most clusters contain less than 5 records). As the estimates can be sensitive to the construction of the clusters, we apply this procedure for cluster radii of $l \in \{100, 200, ..., 1000\}$ km and for the minimum required number of records in a cluster of

$n \in \{2, 3\}$.





The median SNR over all sensitivity experiments is $1.6 \pm 0.3$ $(1\sigma)$ and the median decorrelation length is $1289 \pm 212$ yrs. When we decompose the SST variability of each proxy record into a signal and a noise component according to SNR=1.6, the mean noise level across all records is $0.9 \pm 0.6$ K. This estimate is consistent with an estimate of $0.6 - 1.3$ K by Tierney et al. (2020) in a data assimilation framework characterizing LGM-to-Holocene anomalies. Our estimate is slightly higher than the

SNR of 1.0 employed in the LGM climate field reconstruction by Paul et al. (2021).

### 3.3   Pseudo-proxy experiments

We use PPEs for three purposes: (i) to demonstrate the main features in the simulations that are captured by the model-data comparison algorithm; (ii) to diagnose how much model-data comparison results depend on limited temporal resolution, chronological uncertainties, and the magnitude and temporal autocorrelation structure of non-climatic noise; and (iii) to inves-

tigate how sensitive results are when noise magnitude and temporal autocorrelation structure in the PSM are different from their optimal values. Note that the difference between (ii) and (iii) is that (ii) is motivated by quantifiable limitations and uncertainties of reconstructions, while (iii) targets specifically the fact that the employed PSM is just an approximation of reality and its optimal parameters are unknown.

In PPEs, the underlying climate evolution is given by a reference simulation. For each proxy record, the PSM from Sect. 3.1.1

is applied to the reference simulation with $N = 1$ to generate a single realization of forward-modeled proxy time series with a randomly selected iteration of the age-depth model and one realization of the non-climatic noise process. As this realization mimics the properties of the SST reconstructions, we call it pseudo-proxies. We simulate pseudo-proxies at the locations and with the time axes and chronological uncertainties of the 74 selected proxy records from Sect. 2.2. Then, the algorithm from Sect. 3.1 is employed to compute the deviations between $N = 100$ realizations of forward-modeled proxy time series derived

from each simulation and the pseudo-proxies.

For (i), we use an example PPE with a subset of simulations to illustrate how simulations' characteristics influence their ranking by our algorithm. We use MPI_Glac1D_P3 as reference simulation and PSM parameters given by the estimates from Sect. 3.2 (SNR=1.6, decorrelation length = 1289 yrs). For the PPE, we select simulations that differ from the reference simulations in boundary conditions (MPI_Ice6G_P2_noMWF, TraCE-ALL), parameter configuration (MPI_Glac1D_PTK), and

employed climate model (TraCE-ALL). Additionally, two idealized modifications of MPI_Glac1D_P3, which are shifted in time by 2 kyr in either direction (MPI_Glac1D_P3-2k, MPI_Glac1D_P3+2k), show the effects of a timing mismatch in the deglacial temperature evolution on the model-data comparison results (Fig. 4a).

For (ii) and (iii), we perform two sets of PPEs (Table 3). In the first set, we assume that the PSM structure (noise magnitude and type) is known but we systematically vary the SNR of the records from very low (SNR=1/4) to very high (SNR=16) and

include PPEs without additive noise process (SNR=Inf). We further vary the noise type between white noise (no autocorrelation), an AR1 process with a decorrelation length of 1 kyr, and a self-similar process following a power-law distribution with exponent one (red noise). Using all ten transient simulations as reference simulations to avoid spurious results from selecting a specific reference simulation, we perform in total 240 PPEs (8 SNRs, 3 noise types, 10 reference simulations).



In the second set, the PSM structure used for generating the forward-modeled proxy time series employed in the model-
data comparison algorithm deviates from the one selected to simulate the pseudo-proxies, thus imitating the case where the
PSM structure is uncertain. For each of the ten reference simulations, we draw a realization of pseudo-proxies with AR1
noise (SNR=2, decorrelation length = 1 kyr). For each pseudo-proxy realization, we first apply the model-data comparison
algorithm with varying SNRs in the PSM (SNR=1/4 to SNR=16 and SNR=Inf) but the same autocorrelation structure as in the
construction of the pseudo-proxies. Then, we apply the model-data algorithm with varying autocorrelation structure (white,
AR1, and power-law noise) but the same SNR as in the construction of the pseudo-proxies.

Whether a certain IQD corresponds to an acceptable agreement between a simulation and a reconstruction is a subjective
choice. Moreover, because the IQD uses the probability distribution of the forward-modeled proxy time series, the absolute
value of the IQD depends on the specification of the PSM. For example, a higher SNR results in a lower spread of the forward-
modeled proxy time series created from the same simulation, such that the IQD for a high SNR will differ from the IQD
for a low SNR, even if the simulated and reconstructed SST time series are the same. Therefore, we focus on the ability of
the algorithm to reliably discriminate between simulations, i.e. determining whether simulation $A$ is closer to reality than
simulation $B$. In PPEs, we can compute the 'ground truth deviation' between a simulation and the reference climate history
that was used to construct the pseudo-proxies. We choose the mean absolute deviation from the reference simulation at the
locations of the proxy records as ground truth deviation because the IQD reduces to the mean absolute difference in the
absence of uncertainties. Then, we compute a reference ranking by sorting the simulations according to their ground truth
deviations. Similarly, we can rank the simulations according to the IQDs between the forward-modeled proxy time series and
the pseudo-proxies, which is the ranking that would be obtained in a real-world model-data comparison situation in which only
the pseudo-proxies are known but not the underlying reference climate history. We call this the pseudo-proxy ranking.

Finally, we compare the reference ranking with the pseudo-proxy ranking. If the model-data comparison algorithm discrim-
inated perfectly between simulations, the reference ranking and pseudo-proxy ranking would be identical. However, due to
reconstruction uncertainties and limitations, this will not always be the case. To quantify the similarity of the two rankings, we
introduce a measure called the 'fraction of pairwise reversed rankings' (FPRR). This measure is based on pairwise comparisons
of the rankings of simulations: if simulation $A$ ranks higher than simulation $B$ in the reference ranking, but ranks lower in the
pseudo-proxy ranking, we say that the ranking of the two simulations is reversed in the pseudo-proxy ranking, i.e. the two
simulations are erroneously ranked by the model-data comparison algorithm. We assign $1$ to the pairwise comparison if the
ranking is reversed and $0$ if it is not reversed. We compare the rankings for all pairs of simulations and define the FPRR as the
mean of all pairwise comparisons. The FPRR is $0$ when the pseudo-proxy and reference rankings are equal and it is $1$ if the two
rankings are exactly reversed. The expected value for a random ranking of simulations is 0.5, which means that an FPRR below
0.5 indicates a better-than-random ranking. We focus on two aspects of the simulations' rankings: (i) the reliability of rankings,
i.e. the expected probability of erroneously ranking simulations which we define as the median IQD in a set of PPEs with the
same PSM parameters; (ii) the robustness of rankings, i.e. how much the probability of an erroneous ranking depends on the
reference climate history and the realization of non-climatic processes in the pseudo-proxies. Robustness is quantified by the



spread of the IQD in a set of PPEs with the same PSM parameters and can be interpreted as a measure for the predictability of the reliability of model-data comparison results.

## 4  Results

We start this section with an example PPE that demonstrates the characteristics of the model-data comparison algorithm. Then, we use the PPE framework to systematically assess the dependency of model-data comparison results on uncertainties and limitations of SST reconstructions, and the robustness of the algorithm when PSM structures are uncertain. Finally, we demonstrate our algorithm in a real-world setting by quantifying the deviations between deglacial simulations and SST reconstructions.

### 4.1  Pseudo-proxy experiments

#### 4.1.1  Exemplifying pseudo-proxy experiment

As described in Sect. 3.3, we use an example PPE with MPI_Glac1D_P3 as reference simulation to demonstrate how a simulation's characteristics influence their ranking by our algorithm. The globally averaged ground truth deviations, i.e. IQDs between simulations and the reference simulation at the proxy locations with a regular temporal resolution, no chronological uncertainties, and no non-climatic noise, are shown in Fig. 4b,d, and the IQDs from the comparison between forward-modeled proxy time series and pseudo-proxies in Fig. 4c,e. For all four components of the deglacial temperature evolution (orbital magnitudes, millennial magnitudes, orbital patterns, and millennial patterns), the spread between IQDs corresponding to different simulations are smaller in the PPE (Fig. 4c,e) than in the ground truth deviations (Fig. 4b,d). This shows that in the presence of uncertainties, the forward-modeled proxy time series constructed from different simulations are harder to distinguish than the simulations in the uncertainty-free ground truth. However, the pseudo-proxy ranking mostly preserves the reference ranking (see Sect. 3.3 for definition), which demonstrates the ability of the algorithm to still discriminate correctly between simulations in the presence of reconstruction limitations and uncertainties.

Comparing the IQDs with simulated GMSAT anomalies (Fig. 4a), we see that the orbital magnitude IQD rankings follow the differences in the magnitude of deglacial warming compared to the reference simulation. For millennial magnitude IQDs, meltwater fluxes have a strong influence. MPI_Ice6G_P2_noMWF, in which no meltwater flux is applied, deviates substantially from the reference simulation. The varying spatial structure of millennial magnitudes due to the different meltwater history between TraCE-ALL and MPI_Glac1D_P3 seems to be exaggerated in the pseudo-proxy IQDs. This leads to TraCE-ALL having a higher millennial magnitude IQD than MPI_Ice6G_P2_noMWF in the PPE but not in the ground truth.

The orbital pattern IQDs do not vary strongly between the MPI-ESM simulations, which all feature similar warming trends. In contrast, deglacial warming starts later and is more abrupt in TraCE-ALL, which results in a larger orbital pattern IQD. The difference in the meltwater histories is reflected in the millennial pattern IQDs: MPI_Glac1D_P3 and MPI_Glac1D_PTK feature smaller IQDs than MPI_Ice6G_P2_noMWF, which does not exhibit pronounced millennial-scale fluctuations. The millennial pattern IQD is largest in TraCE-ALL, where a strong fluctuation around 14.5 ka is of opposite sign to MPI_Glac1D_P3.





In the reference rankings as well as the PPE, the time-shifted versions of MPI_Glac1D_P3 are very similar to the reference
simulation in the magnitude components (Fig. 4b,c). This is because the magnitude of orbital and millennial variations changes
little under time shifts. In contrast, time-shifted versions deviate substantially from the reference simulation in the temporal
patterns (Fig. 4d,e) because the timing of the start and end of the deglacial warming as well as the millennial-scale fluctuations
differs from the reference simulation. This shows that the magnitude IQDs are insensitive to differences in the timing of events
whereas timing differences show pronounced in the pattern IQDs.

**4.1.2 Dependency of simulation rankings on non-climatic noise level**

We analyze the first set of 240 PPEs (see Sect. 3.3, set 1 in Table 3) by aggregating them according to the employed SNR
and compare the respective FPRRs for three averaging scales: globally, zonally, and locally (Fig. 5). For all averaging scales,
FPRRs increase for lower SNRs, i.e. pseudo-proxy rankings deviate more from the reference ranking for higher noise levels.
However, even for the highest considered noise levels, the FPRRs are rarely above 0.5. Thus, there is almost always enough
information of the underlying signal preserved to obtain a better than random ranking. There is no threshold behavior, but a
steady FRPRR increase for lower SNRs. The increasing FPRRs for lower SNRs are expected since higher non-climatic noise
levels make it harder to distinguish simulations.

On average, rankings of orbital magnitudes differ least from the reference rankings, followed by orbital patterns, and millen-
nial patterns. Millennial magnitude rankings are the least reliable under non-climatic noise. More reliable orbital than millennial
rankings are expected because temperature variations are larger on orbital than millennial timescales whereas the noise level
does not increase by the same rate on longer timescales. Median FPRRs mostly increase for decreasing spatial averaging scales,
i.e. the reliability of rankings decreases from globally to locally averaged IQDs. The spread of FPRRs over the PPEs with the
same SNR tends to increase with higher noise level and smaller spatial averaging scale, too. Thus, model-data comparison
results are not just less reliable but also less robust for higher noise levels and smaller averaging scales (see also Sect. 3.3).
For our SNR estimates from Sect. 3.2, the PPE results suggest below 10% expected erroneous simulation rankings for orbital
magnitudes and patterns and 10-20% for millennial patterns and magnitudes.

**4.1.3 Stability of simulation rankings for uncertain proxy system models**

In reality, the magnitude and temporal structure of non-climatic processes is uncertain. Therefore, we test how robust model-
data comparison results are when either the SNR or the temporal autocorrelation structure in the forward-modeled proxy time
series differs from the values selected to construct the pseudo-proxies (see set 2 in Table 3). In Fig. 6, we show the FPRR
for over- or under-estimated SNRs and for over- (power-law) or under-estimated (white noise) temporal persistence of non-
climatic processes. We find small influences from moderately (factor 2 to 4) over- or underestimating the SNR. Substantial
differences from the results for the true SNR only occur for strong deviations (larger than factor 4) from the true SNR or
when non-climatic processes are neglected entirely (SNR=Inf). For all averaging scales and all four components, the effects of
misspecified temporal autocorrelation structures are negligible.



The FPRR medians and spreads for orbital magnitudes and patterns vary very little for over- or underestimated SNRs across the whole range of SNRs. Thus, correctly estimating SNRs or the temporal structure of the autocorrelation has very little influence on the reliability and robustness of orbital-scale IQDs. For millennial patterns, results are very stable as long as non-climatic noise is not completely neglected (SNR=Inf). For SNR=Inf, medians and spreads of FPRRs both increase, but the medians are still below the 95th FPRR percentile for the correct SNR. The influence of misspecified SNRs is largest for millennial magnitudes, where we find two opposing trends. On the one hand, the median FPRR stays relatively constant for overestimated SNRs but tends to increase for underestimated SNRs. On the other hand, the spread varies little for underestimated SNRs but increases for overestimated SNRs. This suggests that the reliability for millennial magnitudes decreases when the SNR is underestimated whereas the robustness is lower when the SNR is overestimated. For millennial magnitudes, neglecting non-climatic noise entirely reduces the reliability more for global averages than on smaller spatial scales (see also Sect. 5.1).

In summary, within the SNR uncertainty range from Sect. 3.2 (factor of $\sim 2$), the reliability and robustness of the algorithm seem to be very little affected by misspecified SNRs. Substantial reductions of median or spread of FPRR distributions only occur for millennial magnitudes when the SNR is strongly over- or underestimated (factor 4 and larger) and for millennial patterns when non-climatic noise is neglected entirely. The effect of under- or overestimating the temporal persistence of non-climatic noise is negligible in our PPEs, supporting the decision to choose an AR1 process in Sect. 3.2 instead of trying to estimate the structure of the temporal autocorrelation function.

## 4.2 Comparison of simulations against SST reconstructions

Next, we quantify the deviations between forward-modeled proxy time series derived from the ten deglacial simulations (Sect. 2.1) and the 74 selected SST records (Sect. 2.2). We employ a PSM with an AR1 non-climatic noise process and vary the SNR between 1.1 and 2.2 and the decorrelation length between 865 yrs and 1712 yrs (Sect. 3.2). We study globally and regionally averaged IQDs for the Southern Hemisphere extratropics (n=10 proxy records), the Tropics (n=44), the extratropical North Atlantic (n=13), and the extratropical North Pacific (n=7) (see Fig. 1). We select these regions based on detected inter-regional dissimilarities of the deglacial temperature evolution in an initial visual inspection of reconstructions and simulations. All regions contain more than five records and thus we expect the results to benefit from the spatial averaging effect found in the PPEs. Fig. 7 shows the IQDs for all four components of the deglacial temperature evolution, simulations, and regions.

### 4.2.1 Orbital-scale variations

For orbital magnitudes, MPI_Glac1D_P3, MPI_Glac1D_PTK, and TraCE-ALL feature the smallest IQDs between forward-modeled proxy time series and reconstructions in the global average (Fig. 7a). Among these three simulations, MPI_Glac1D_PTK and TraCE-ALL warm by $\sim 4$ K during the deglaciation (see Fig. 1) and deviate less from the reconstructions than other simulations in the Southern Hemisphere and Tropics. Meanwhile, MPI_Glac1D_P3 has the strongest deglacial warming among the simulations and deviates significantly less from the reconstruction in the North Atlantic than all other simulations. In the global average, these regionally varying agreements compensate which shows that GMSAT anomalies alone are not sufficient





to explain the rankings. TraCE-ORB and FAMOUS forward-modeled proxy time series, which warm the least among the en-
semble, deviate most from the reconstructions. In the Tropics and Southern Hemisphere, forward-modeled proxy time series
with median orbital magnitudes around 1 K tend to deviate least from the reconstructions (Fig. 8a). In the North Atlantic, no
simulation matches the high orbital magnitudes of the reconstructions (Fig. 8a). Here, the simulation with the highest magni-
tude (MPI_Glac1D_P3) features the lowest IQDs. In the North Pacific, orbital magnitudes are much smaller than in the North
Atlantic in reconstructions as well as all simulations, and IQDs are relatively similar IQDs for all simulations.

Turning to orbital patterns, the globally averaged IQD differences between simulations are relatively small, except for a
higher mean IQD of TraCE-ORB (Fig. 7b). This can be explained by TraCE-ORB being the only simulation without a clear
warming trend in the Southern Hemisphere (not shown). In the North Atlantic, two distinct regional clusters appear in the
reconstructions (Fig. 9a,c): along the Iberian Margin and in the Mediterranean Sea (denoted Mediterranean North Atlantic,
see Fig. 1b), the lowest SSTs occur during Heinrich Stadial 1 ($\sim$ 17 ka), followed by two strong warming phases, which
are interrupted by a warming hiatus during the Younger Dryas ($\sim$ 12 ka). Meanwhile, warming is more monotonic in the
Subpolar North Atlantic (see Fig. 1b for a definition of the region). In contrast to the reconstructions, the orbital patterns are
very similar between those two subregions of the North Atlantic in all of the simulations (Fig. 9a,c). Due to the differences
between Subpolar and Mediterranean North Atlantic in the reconstructions, the lowest orbital pattern IQDs in the North Atlantic
occur in MPI_Ice6G_P2_noMW, TraCE-GHG, and MPI_Glac1D_P3, which feature a smoother orbital pattern with weaker
interruptions of the warming trend than other simulations. Among all examined regions, the highest orbital pattern IQDs occur
in the North Pacific, where inter-model differences of orbital patterns are also the largest (Fig. 9e). Here, TraCE-ALL and
FAMOUS have the lowest IQDs as these are the only simulations that somewhat resemble the pattern in the reconstructions
with increasing temperature until $\sim$14 ka and subsequent cooling into the Holocene.

### 4.2.2 Millennial-scale variations

Millennial magnitude IQDs exhibit small differences between the simulations containing meltwater-induced abrupt events
when averaged globally as well as in the Southern Hemisphere extratropics and in the Tropics (Fig. 7c). In the global aver-
age, the simulations with the weakest millennial-scale variability (Fig. 8b) agree least with the reconstructions. The highest
millennial magnitudes in reconstructions and simulations occur in the North Atlantic (Fig. 8b). Here, two simulations with
medium millennial magnitudes, TraCE-ALL and MPI_Glac1D_PTK, have the smallest IQDs, whereas largest deviations from
the reconstructions occur for simulations without meltwater input. Compared to the North Atlantic, millennial-scale variations
are weaker in the North Pacific in reconstructions and simulations and IQDs are more similar between simulations.

Turning to millennial patterns, two simulations without distinct millennial-scale variations feature the lowest globally-
averaged IQDs (TraCE-GHG, MPI_Ice6G_P2_noMWF, Fig. 7d). This is because no single simulation with distinct millennial-
scale variations reproduces the reconstructed millennial patterns effectively in all regions. The agreement between simulations
and reconstructions even differs within the North Atlantic and between North Atlantic and North Pacific (Fig. 9). Here, the
meltwater fluxes extracted from the ice sheet reconstructions through dynamic river routing in the MPI-ESM simulations lead
to abrupt millennial-scale temperature variations that do not align with the reconstructions. TraCE-ALL matches the millennial-




scale variability pattern in the Mediterranean North Atlantic and therefore features the smallest IQDs in this area (Fig. 9b). However, it deviates strongly from the reconstructions in the Subpolar North Atlantic (Fig. 9d) and North Pacific (Fig. 9f).

## 5 Discussion

Our study is a first step towards quantitative spatio-temporal model-data comparison for transient simulations of past climate transitions, as demonstrated here for the LD. In this section, we explore reasons for the PPE results and their implications. Then, we discuss the agreement between transient simulations of the LD and SST reconstructions, provide ideas for testing potential reasons for disagreements, and suggest improvements for future applications.

### 5.1 Reliability and robustness of the model-data comparison algorithm

The systematic PPEs show that the reliability and robustness of simulation rankings decrease with increasing noise levels. This result is not surprising as higher noise levels make it harder to identify the underlying temperature signal. The effect can be reduced by spatially averaging IQDs over multiple records. As we assume the non-climatic noise to be independent between records, averaging over IQDs from multiple records reduces the influence of the noise and thus effectively enhances the SNR. If modulations of the temperature signal were not independent between records in reality, the improvement when averaging IQDs from multiple records would be weakened.

Rankings for orbital-scale variations are more reliable and robust than for millennial-scale variations due to comparably smaller distortion by non-climatic noise. That orbital magnitude rankings tend to be more reliable and robust than orbital pattern rankings could be due to relatively subtle differences between simulations in the timing and shape of the deglacial warming trend compared to easier to identify differences in the magnitude of deglacial warming. On the other hand, we attribute more reliable and robust millennial pattern than magnitude rankings to the differing effects of non-climatic noise on these two components. Millennial patterns of simulations are often still distinguishable based on their most pronounced fluctuations that are comparatively less distorted by non-climatic noise. Meanwhile, non-climatic noise enhances the magnitude of reconstructed millennial-scale variations (in our PSM proportional to the variability of the simulation at a given location) and thus has a systematic effect on millennial magnitudes which can further diminish the reliability rankings.

If the assumed SNR in the model-data comparison is not strongly over- or underestimated (factor 4 and more), results remain reliable. Using explicitly conservative SNR values is not safeguarding from erroneous rankings as strongly underestimating SNRs reduces the reliability whereas strongly overestimating SNRs reduces the robustness of rankings. Incorrect specifications of the temporal autocorrelation structure of non-climatic processes have a negligible effect in our PPEs. This rather unexpected result might be due to the relatively short time period of investigation (16 kyr) compared to the timescales we study. This hypothesis could be tested in future work by repeating the experiments for longer periods. Entirely neglecting existing non-climatic processes leads to less robust and reliable rankings for millennial-scale variations. On the one hand, this can be explained by non-climatic variations in reconstructions being interpreted as climate signals for SNR=Inf, such that rankings depend more on the unknown realization of non-climatic processes. On the other hand, underestimating millennial-scale



variations by neglecting variability-enhancing processes can systematically distort millennial magnitude rankings. This effect is strongest for global averages.

Taken together, the PPE results suggest that the reliability and robustness of model-data comparison results can be improved the most by increasing the SNR. In contrast, reducing the uncertainty of the SNR or improving the specification of the temporal autocorrelation structures will barely improve rankings. A doubling of the SNR typically reduces erroneous rankings by 1-3

percentage points. Thus, incremental improvements of the SNR, for example through process-based modeling of modulations of the recorded climate signal, will only have a small effect on the reliability of rankings. PPEs with SNR=Inf typically still have 5-10% erroneous rankings for regionally averaged IQDs. This percentage could be reduced by more precise chronologies and higher temporal resolutions of records. Comparing global, zonal, and local estimates suggests that significantly improved reliability can also be achieved by increasing the number of proxy records and thus averaging over more records in regional

averages, as long as non-climatic contributions are not strongly correlated between records.

### 5.2    Agreement of SST reconstructions and deglacial simulations

The diversity of the simulations in terms of employed climate model and boundary conditions can provide insights into their importance for model-data disagreements. Comparing the MPI-ESM and CCSM3 simulations that employ orbital, GHG, and ice sheet forcing, we find no systematic differences between the two climate models. In particular, TraCE-ALL is mostly

within the IQD spread of the six MPI-ESM simulations. This indicates that parameter configurations and employed ice sheet reconstructions are more important to explain regionally varying model-data agreement than the structural differences between the two models. For example, discrepancies in the response of MPI-ESM to the GLAC-1D and ICE-6G reconstructions result in a significantly higher orbital-scale agreement of the simulation employing GLAC-1D with the reconstructions in the North Atlantic (for a detailed analysis of these differences see Kapsch et al., 2022). Meanwhile, we find a systematically larger orbital

magnitude mismatch between FAMOUS and the reconstructions compared to all MPI-ESM simulations and TraCE-ALL. The larger mismatch can be explained by a lower deglacial warming in FAMOUS, but sensitivity experiments would be needed to test if this is a structural characteristic of FAMOUS or a result of choices in the simulation design such as the acceleration in the forcing, which can delay global warming, or the absence of transiently changing land-sea masks and Southern Hemisphere ice sheets.

The simulation with transient changes of orbital parameters only (TraCE-ORB) deviates significantly more from the reconstructions than all other simulations for orbital magnitudes, orbital patterns, and millennial magnitudes. This is due to too small magnitudes of variability in most regions and the absence of a deglacial warming trend in the Southern Hemisphere when GHG and ice sheet changes are neglected. In contrast, the neglected orbital and ice sheet forcing in TraCE-GHG do not lead to clearly higher disagreements for orbital-scale variability and millennial patterns. The latter could again be due to the

insufficient sampling of ice sheet reconstruction uncertainties by the simulation ensemble. Meanwhile, the absence of ice sheet forcing is degrading results strongly for millennial magnitudes. More generally, all simulations with meltwater input show a better agreement with reconstructions for millennial magnitudes than those without meltwater input. The improved agreement originates mostly from a higher millennial-scale variability in the North Atlantic, where the meltwater-induced variability is



the strongest. Meanwhile, two of the simulations without meltwater fluxes have the smallest millennial pattern disagreement in
the global average, which suggests that none of the employed meltwater schemes leads to a temporal pattern of millennial-scale
variability (e.g. timing, direction, and length of abrupt warming/cooling events) that is globally consistent with the reconstruc-
tions. As ice sheet reconstructions are highly uncertain (Stokes et al., 2015; Abe-Ouchi et al., 2015; Ivanovic et al., 2016), our
results are insufficient to determine whether insufficient sampling of forcing uncertainties, the prescribed input location of melt-
water in the simulations, the simulated response to meltwater fluxes, or mismatches in the meltwater-independent variability
are mainly responsible for the millennial pattern disagreements.

On the whole, we find that no simulation ranks among the simulations with the smallest deviation from the reconstructions
across all four components and considered regions. Examples of regionally varying mismatches between simulations, which
compensate in global averages, are found for all four components of the deglacial temperature evolution (see Sect. 4.2). These
compensations occur because simulations with higher variability than others have a higher variability in almost all regions (Fig.
8). Additionally, simulations tend to have similar temporal patterns at least within each hemisphere (Fig. 9). In contrast, the
reconstructed variability magnitudes are most similar to the simulations with the highest variability in some regions, but closer
to those with low variability in others. Similarly, the reconstructed variability patterns vary more between and within ocean
basins than in the simulations. Therefore, we attribute the absence of a simulation with consistently high agreement relative
to the others to more regionally confined variability magnitudes and patterns in reconstructions than in simulations. In other
words, the reconstructed spatial variability of the deglacial temperature evolution is higher than in all considered simulations.
For the North Atlantic, the differences in the reconstructed deglacial temperature evolution between the Mediterranean and the
Subpolar North Atlantic found in this study are consistent with a recent synthesis by Pedro et al. (2022).

This mismatch in the spatio-temporal variability structure could be caused by uncertainties in ice sheet reconstructions,
shortcomings of the employed models, or temperature reconstruction characteristics that vary between regions. The role of
systematic reconstruction deviations from mean annual SST can be assessed by integrating process-based PSMs (e.g. Dolman
and Laepple, 2018; Kretschmer et al., 2018; Osman et al., 2021) into our algorithm in future work. This could disentangle
the importance of different processes occurring during the recording, archiving, and measuring of the sensors, e.g. recording
season and depth preferences, confounding environmental variables, and bioturbation. The locations of proxy records are biased
towards coastal regions and, for some regions, our results rely on records clustered in small areas. This could reduce the model-
data agreement if the resolution of models was insufficient for an accurate simulation of zonal temperature heterogeneity, e.g.
due to coastal upwelling or deficiencies in the simulation of gyre circulations and air-sea interactions (Seager et al., 2003;
Kwon et al., 2010; Ma et al., 2016; Judd et al., 2020; Paul et al., 2021). As higher resolution simulations of the deglaciation
are currently precluded by computational limitations, including more proxy data and physically-motivated downscaling of
simulation output could help to test this explanation. Finally, the reconstructed meltwater peaks could be too high or the
models' responses to them too strong, leading to a spatially too homogeneous SST response (He and Clark, 2022). Insights
on this potential explanation could be gained from coupled atmosphere-ocean-ice sheet simulations (Ziemen et al., 2019) or
replacing local meltwater input by freshwater fingerprints obtained from eddy-resolving ocean models (Love et al., 2021).





As the PPEs and the real-world application have shown, the pattern IQDs are sensitive to the timing of timescale-dependent temperature fluctuations. Therefore, they are only meaningful if the goal of a simulation is to reproduce a specific succession of variations observed in reconstructions. In the presence of uncertain meltwater fluxes and for simulations with spontaneous millennial-scale fluctuations (Obase and Abe-Ouchi, 2019; Vettoretti et al., 2022), the magnitude IQDs, which are insensitive to the timing of fluctuations, could be combined with a more insightful measure for temporal patterns. Such a measure could be based on the similarity of spatial relationships in reconstructed and forward-modeled proxy time series (e.g. Adam et al., 2021).

Applications of our model-data algorithm are not restricted to SST reconstructions during the last deglaciation. With new syntheses becoming available (Herzschuh et al., 2022), an extension to terrestrial temperature records can be attempted. Moreover, other periods with climate transitions and periods with changing background conditions can be assessed, as long as a sufficient number of proxy records with absolute chronologies are available. Targets could for example be the penultimate deglaciation, the glacial inception, or the last glacial cycle.

## 6 Conclusions

We present a new approach for the spatio-temporal comparison of reconstructed and simulated deglacial temperature evolutions. To avoid the need to reconstruct gridded or regional mean temperatures from sparse and uncertain proxy data, the algorithm applies proxy system models to simulation output and quantifies the deviation between the resulting forward-modeled proxy time series and temperature reconstructions. We assess the reliability and robustness of the algorithm in pseudo-proxy experiments. For signal-to-noise ratios as estimated from a database of sea surface temperature reconstructions, the expected rate of simulation pairs that are ranked erroneously compared to the underlying ground truth is less than 10% for magnitudes and temporal patterns of orbital-scale variations and 10-20% for millennial-scale magnitudes and patterns, when deviations are regionally averaged. The quality of rankings is barely influenced by uncertainties in proxy system model parameters. The reliability and robustness of rankings could be improved most by including more data and increasing the signal-to-noise ratio.

Comparing ten transient simulations of the last deglaciation with a global compilation of sea surface temperature reconstructions, we demonstrate that the algorithm provides insights into the importance of model differences and boundary conditions for explaining mismatches between simulations and reconstructions. The ranking of the simulations differs substantially between the considered regions and timescales and no simulation features a consistently high agreement with the reconstructions. We attribute this result to greater differences between and within ocean basins in reconstructions than in simulations. The mismatch could originate from uncertainties in boundary conditions, shortcomings of the employed climate models, or reconstruction characteristics that vary between regions. Further analyses are required to disentangle these potential explanations. Beyond quantifying disagreements between a given simulation and a database of reconstructions, our algorithm can be used for model tuning, testing the influence of uncertain boundary conditions, and understanding influences of non-climatic processes on model-data mismatches.



*Code and data availability.* R code to reproduce the results and plots of this study is available at https://doi.org/10.5281/zenodo.7924111. The PalMod 130k marine paleoclimate data synthesis v1.1.1 is available at Jonkers et al. (2023). MPI-ESM simulation data was processed and provided by Marie Kapsch, Uwe Mikolajewicz, and Thomas Kleinen. Output from the MPI_Glac1D_P3, MPI_Ice6G_P3, MPI_Ice6G_P2, and MPI_Glac1D_PTK simulations is also available at https://esgf-data.dkrz.de/projects/palmod (last accessed: 28.02.2023). TraCE data was obtained from https://www.earthsystemgrid.org/project/trace.html (last accessed: 28.02.2023), and FAMOUS data was obtained from https:
//data.ceda.ac.uk/badc/quest/data/quaternaryq/famous_glacial_cycle (last accessed: 28.02.2023). More information on access to simulation output is available in the respective original publications (Kapsch et al., 2022; Kleinen et al., 2022; Liu et al., 2009; Smith and Gregory, 2012).

*Author contributions.* NW, KR, and HA designed the study with input from OB, LJ, and AP. JPB, LJ, MK, TK, UM, NW, and EZ processed
the data. NW implemented and ran the model-data comparison algorithm. All authors discussed the results. NW wrote the manuscript with input from KR and HA. All authors commented on earlier versions of the manuscript and approved the final manuscript.

*Competing interests.* The authors declare that they have no competing interests.

*Acknowledgements.* This work originated from a workshop organized by OB and funded by Helmholtz-Zentrum Hereon and PalMod. NW, EZ, and KR acknowledge funding by the Deutsche Forschungsgemeinschaft (DFG, German Research Foundation), project no. 395588486. HA, JPB, OB, LJ, MK, TK, UM, AP, and NW acknowledge funding from the German Federal Ministry of Education and Research (BMBF)
within the Research for Sustainability initiative (FONA; https://www.fona.de/, last access: 10 November 2022) through the PalMod project, grant nos. (FKZ): 01LP1926C (JPB, NW), 01LP1509A (OB), 01LP1926B (OB), 01LP1922A (LJ), 01LP1504C (MK), 01LP1917B (MK), 01LP1921A (TK), 01LP1915C (UM), 01LP1511D (AP). All Max Planck Institute for Meteorology Earth System Model simulations were performed at the German Climate Computing Center (DKRZ). We thank Andrew Dolman for his helpful comments on a previous version of the manuscript.



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

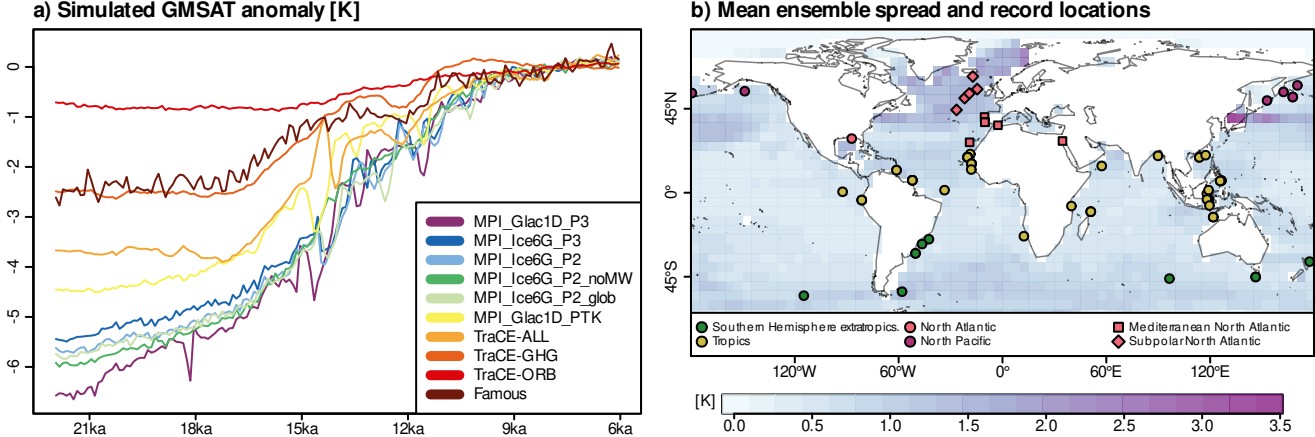

**Figure 1.** (a) GMSAT anomalies of the transient simulation ensemble members. Anomalies were computed with respect to the mean in the window 9 ka to 6 ka. (b) Locations of SST reconstruction records employed in the model-data comparison (dots) and simulation ensemble spread as measured by the standard deviation at each location and time step, averaged over all time steps (colors in the background). The colors of the dots indicate the regions considered in Sect. 4.2 and the shape of the dots in the North Atlantic mark the records used for the separation into Mediterranean and Subpolar North Atlantic in Sect. 4.2 and Fig. 9. Ocean grid cells are selected based on the ICE-6G history (Peltier et al., 2015).



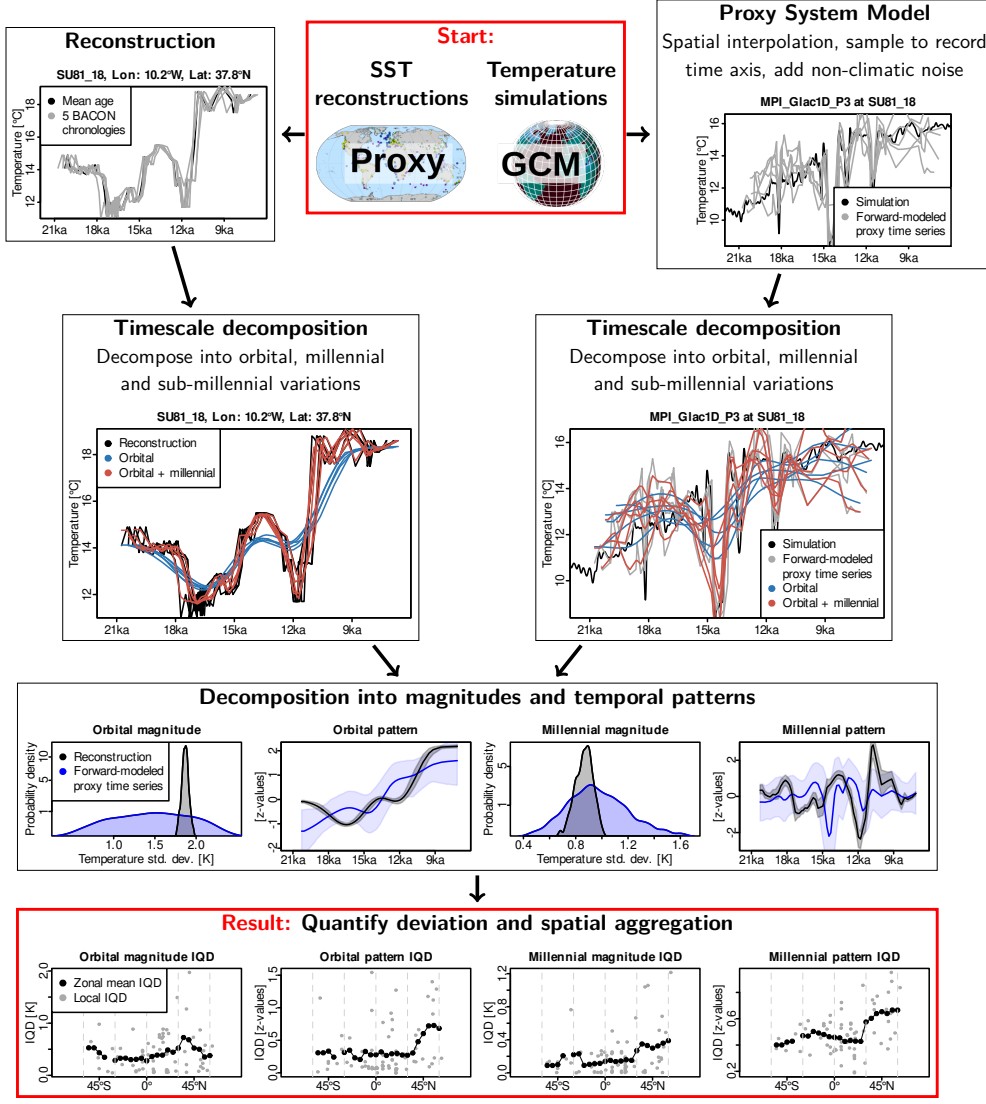

**Figure 2.** Flow chart describing the algorithm presented in this study (see Sect. 3 for details). We start at the top with two sets of data, reconstructed and simulated SSTs. Age uncertainties of the proxy records are quantified using multiple iterations from the age-depth model (top row, left). We apply a proxy system model (PSM) to the simulated SST fields to obtain Monte Carlo (MC) samples of forward-modeled proxy time series (top row, right). For each MC sample, a timescale decomposition is performed to separate orbital- and millennial-scale variations using Gaussian smoothers (second row, left for reconstructions, right for forward-modeled proxy time series). Differences between the MC samples of reconstructions are due to chronological uncertainties, whereas differences in the MC samples of forward-modeled proxy time series result from the stochastic PSM. The orbital- and millennial-scale time series are decomposed into the magnitude and temporal pattern of the variations. This leads to probability distributions for reconstructions and forward-modeled proxy time series (third row). Finally, the integrated quadratic distance (IQD) between the probability distributions of reconstructions and forward-modeled proxy time series is computed for each of the four components (dots in the bottom row) and IQDs are averaged spatially (zonal mean IQDs in the bottom row for all latitudinal bands containing at least five proxy records).



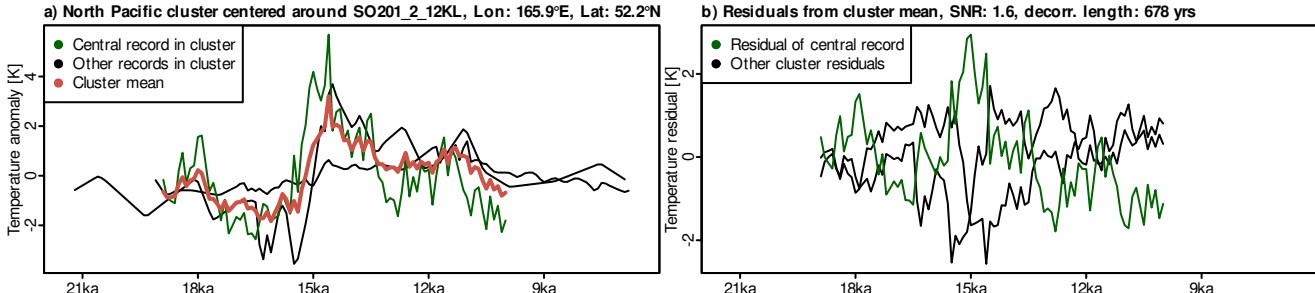

**Figure 3.** Visualization of the PSM parameter estimation as described in Sect. 3.2 for a cluster with 500 km radius and n=3 records in the North Pacific centered around the proxy record SO201_2_12KL. (a) All SST records in the cluster and the corresponding local mean SST reconstruction (red line) with the central record of the cluster in green. (b) Residual deviations from the local mean reconstruction with the central record in green. The SNR and decorrelation length for the central record (green) are given in the caption. SNRs are estimated by comparing the variance of the mean reconstruction (signal) against the variance of the residuals (noise). The decorrelation length of the noise process is estimated from the residual time series.

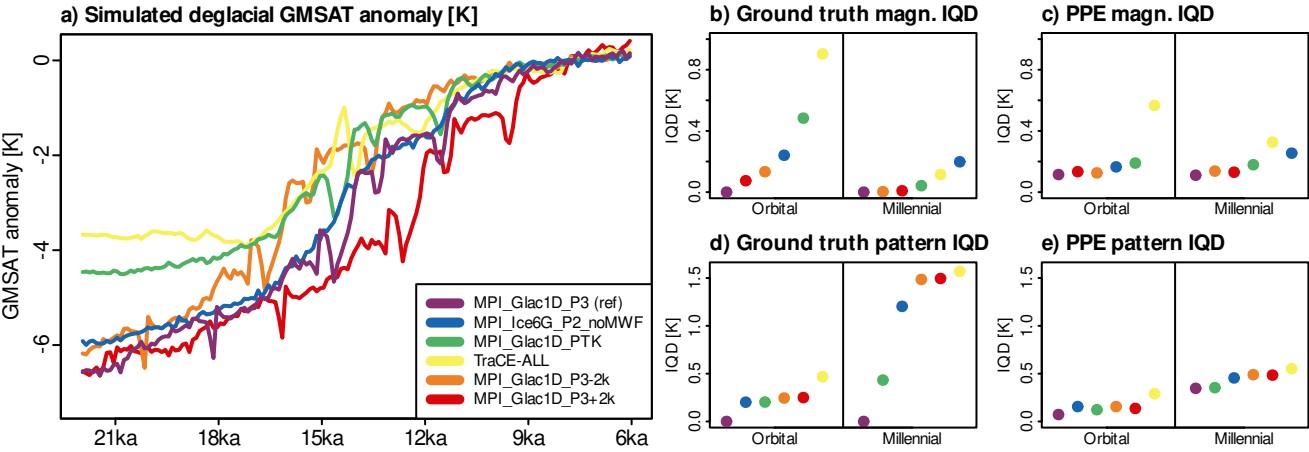

**Figure 4.** Visualisation of the results for a PPE with SNR=1.6, an AR1 noise process with a decorrelation length of 1289 yrs, and MPI_Glac1D_P3 as reference simulation. (a) GMSAT anomalies of the four simulations and the two time-shifted versions of MPI_Glac1D_P3 (anomalies with respect to the mean in the window 9 ka to 6 ka). (b) and (d) show the ground truth magnitude and pattern IQDs (see Sect. 3.3 for details). (c) and (e) are the corresponding deviations between forward-modeled proxy time series and pseudo-proxies constructed from the reference simulation. Note that by definition, the ground truth deviations in (b) and (d) of the reference simulation MPI_Glac1D_P3 from itself are zero.



**Figure 5.** Fraction of pairwise reversed rankings (FPRR, see Sect. 3.3 for definition) of simulations for globally averaged IQDs, zonally averaged IQDs, and IQDs of individual pseudo-proxy records. Shown are FPRRs for (a) orbital-scale magnitudes, (b) millennial-scale magnitudes, (c) orbital-scale temporal patterns, and (d) millennial-scale temporal patterns. Dots depict the medians across all PPEs with a given SNR (n=30 for each SNR). Bars show the spread across PPEs. Darker colors depict the 25th to 75th percentiles, whereas lighter colors depict the 5th to 95th percentiles. SNR=Inf corresponds to PPEs without additive noise process. Dashed horizontal lines indicate FPRRs of 0.05, 0.1, 0.25, and 0.5. FPRRs above 0.5 are worse than expected for a randomized ranking.

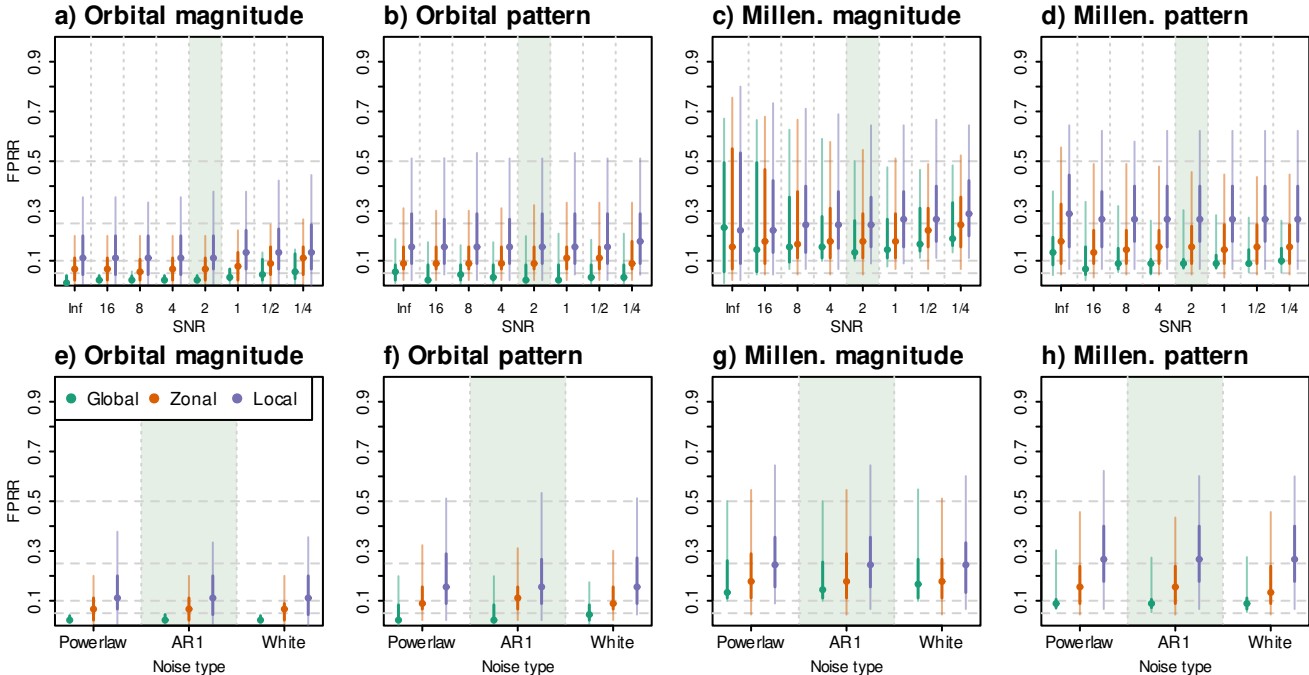

**Figure 6.** Effects of misspecified SNRs and noise types on simulation rankings in PPEs. The reference configuration of the pseudo-proxies in all PPEs is SNR=2 and an AR1 noise with a decorrelation length of 1000 yrs. (a) - (d) show results for the four different components of the deglacial temperature evolution in PPEs with varying SNRs of the forward-modeled proxy time series. (e) - (h) show results for PPEs in which the noise type of the forward-modeled proxy time series varies. Green shaded rectangles indicate the PPEs in which the same noise configuration (SNR=2, AR1 noise) is used for the reference pseudo-proxies and the forward-modeled proxy time series. Dots depict the medians across all PPEs with (a-d) a given SNR (n=10 for each SNR) or (e-h) a given noise type (n=10 for each noise type). Bars show the spread across PPEs. Darker colors depict the 25th to 75th percentiles, whereas lighter colors depict the 5th to 95th percentiles. Dashed horizontal lines indicate FPRRs of 0.05, 0.1, 0.25, and 0.5. FPRRs above 0.5 are worse than expected for a randomized ranking.



**Figure 7.** Global and regional mean IQDs of the ten transient deglacial simulations from the 74 SST reconstruction records. Colored dots show median IQDs for (a) orbital magnitudes, (b) millennial magnitudes, (c) orbital temporal patterns, and (d) millennial temporal patterns. Darker colors depict the 25th to 75th percentiles resulting from varying the uncertain PSM parameters, whereas lighter colors depict the full range of uncertainties from varying the PSM parameters as described in Sect. 4.2. Note that the y-axis ranges are different between the panels.



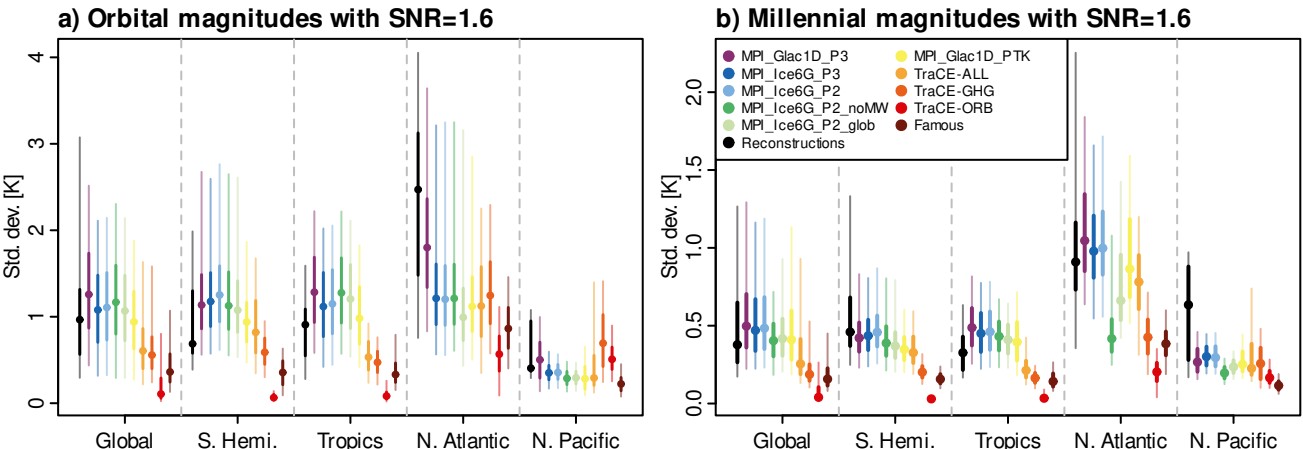

**Figure 8.** Mean absolute magnitudes of timescale-dependent variations of SST reconstructions (black) and forward-modeled proxy time series with the median PSM parameter estimates from Sect. 3.2 (color-coded). Depicted are globally and regionally averaged magnitudes of (a) orbital-scale and (b) millennial-scale variations. Points denote median magnitudes within a region. Darker color bars depict the 25th to 75th percentiles across all records within the respective region, whereas lighter colors depict the 5th and 95th percentile.



**Figure 9.** Regionally stacked temporal patterns of orbital-scale (left column) and millennial-scale (right column) variations for records in (a, b) the Mediterranean North Atlantic, (c, d) the Subpolar North Atlantic, (e, f) the North Pacific (see Fig. 1 for the definition of the regions). Black lines denote the stacked reconstructions, whereas colored lines depict the stacked forward-modeled proxy time series derived from the ten transient simulations. Shaded areas show uncertainties from chronologies and the PSM. The numbers in the legends next to each simulation are the averaged IQDs over all records in the respective stacks.



**Table 1.** Properties of the ten transient simulations of the LD included in the simulation ensemble: name used throughout the manuscript, the employed climate model, whether orbital and GHG forcings were varied transiently or fixed at LGM values, the employed ice sheet reconstructions, how meltwater fluxes were applied (local input through dynamical river routing, local input according to a manually defined scheme, distributed equally across all grid cells, or no meltwater input), and the main reference of the simulation.

| Name | Model | Orbital | GHG | Ice sheets | Meltwater | Reference |
|---|---|---|---|---|---|---|
| MPI_Glac1D_P3 | MPI-ESM-CR | yes | yes | GLAC-1D | river routing | Kapsch et al. (2022) |
| MPI_Ice6G_P3 | MPI-ESM-CR | yes | yes | ICE-6G | river routing | Kapsch et al. (2022) |
| MPI_Ice6G_P2 | MPI-ESM-CR | yes | yes | ICE-6G | river routing | Kapsch et al. (2022) |
| MPI_Ice6G_P2_noMWF | MPI-ESM-CR | yes | yes | ICE-6G | none | Kapsch et al. (2022) |
| MPI_Ice6G_P2_glob | MPI-ESM-CR | yes | yes | ICE-6G | global | Kapsch et al. (2022) |
| MPI_Glac1D_PTK | MPI-ESM-CR | yes | yes | GLAC-1D | river routing | Kleinen et al. (2022) |
| TraCE-ALL | CCSM3 | yes | yes | ICE-5G | local (manual) | Liu et al. (2009) |
| TraCE-GHG | CCSM3 | no | yes | fixed at LGM | none | Liu et al. (2009) |
| TraCE-ORB | CCSM3 | yes | no | fixed at LGM | none | Liu et al. (2009) |
| FAMOUS | FAMOUS | yes | yes | ICE-5G | none | Smith and Gregory (2012) |



Table 2: Information on the 74 proxy records selected for the deglacial model-data comparison.

| ID | Core name | Lon [°E] | Lat [°N] | Ocean basin | Sensor | Reference |
|----|-----------|----------|----------|-------------|--------|-----------|
| 1 | 108_658C | -18.6 | 20.7 | Atlantic | Uk37 | Zhao et al. (1995) |
| 2 | 323_U1340A | -179.5 | 53.4 | Pacific | Uk37 | Schlung et al. (2013) |
| 3 | BOFS31_1K | -20.2 | 19.0 | Atlantic | Plankt. foram. assembl. | Chapman et al. (1996) |
| 4 | BOFS31_1K | -20.2 | 19.0 | Atlantic | Uk37 | Zhao et al. (1995) |
| 5 | BOFS31_1K | -20.2 | 19.0 | Atlantic | MgCa (G. bulloides) | Elderfield and Ganssen (2000) |
| 6 | BOFS31_1K | -20.2 | 19.0 | Atlantic | MgCa (G. inflata) | Elderfield and Ganssen (2000) |
| 7 | BOFS31_1K | -20.2 | 19.0 | Atlantic | MgCa (G. ruber pink) | Elderfield and Ganssen (2000) |
| 8 | BOFS31_1K | -20.2 | 19.0 | Atlantic | MgCa (N. incompta) | Elderfield and Ganssen (2000) |
| 9 | BOFS5K | -21.9 | 50.7 | Atlantic | Plankt. foram. assembl. | Maslin et al. (1995) Vogelsang et al. (2001) |
| 10 | GeoB12615_4 | 39.8 | -7.1 | Indian | MgCa (G. ruber white) | Romahn et al. (2014) |
| 11 | GeoB16224_1 | -52.1 | 6.7 | Atlantic | MgCa (G. ruber white) | Crivellari et al. (2019) |
| 12 | GeoB16224_1 | -52.1 | 6.7 | Atlantic | Plankt. foram. assembl. | Crivellari et al. (2019) |
| 13 | GeoB16224_1 | -52.1 | 6.7 | Atlantic | Uk37 | Crivellari et al. (2019) |
| 14 | GeoB16224_1 | -52.1 | 6.7 | Atlantic | TEX86 | Crivellari et al. (2019) |
| 15 | GeoB16602 | 113.7 | 19.0 | Pacific | Uk37 | Huang et al. (2018) |
| 16 | GeoB16602 | 113.7 | 19.0 | Pacific | MgCa (G. ruber white) | Cheng et al. (2018) |
| 17 | GeoB1711_4 | 12.4 | -23.3 | Atlantic | Uk37 | Kirst et al. (1999) |
| 18 | GeoB5844_2 | 34.7 | 27.7 | Indian | Uk37 | Arz et al. (2003) |
| 19 | GeoB6211_2 | -50.2 | -32.5 | Atlantic | MgCa (G. inflata) | Chiessi et al. (2008) |
| 20 | GeoB6211_2 | -50.2 | -32.5 | Atlantic | MgCa (G. ruber white) | Chiessi et al. (2014, 2015) |
| 21 | GeoB9508_5 | -17.9 | 15.5 | Atlantic | Uk37 | Niedermeyer et al. (2009) |
| 22 | GeoB9508_5 | -17.9 | 15.5 | Atlantic | MgCa (G. ruber pink) | Zarriess et al. (2011) |
| 23 | GeoB9508_5 | -17.9 | 15.5 | Atlantic | MgCa (G. inflata) | Bouimetarhan et al. (2013) |
| 24 | GeoB9508_5 | -17.9 | 15.5 | Atlantic | MgCa (G. bulloides) | Bouimetarhan et al. (2013) |
| 25 | GeoB9526_5 | -18.1 | 12.4 | Atlantic | MgCa (G. ruber pink) | Zarriess et al. (2011) |
| 26 | GIK15612_2 | -26.5 | 44.4 | Atlantic | Plankt. foram. assembl. | Kiefer (1998) |
| 27 | GIK15637_1 | -19.0 | 27.0 | Atlantic | Plankt. foram. assembl. | Kiefer (1998) |
| 28 | GIK17286_1 | 89.9 | 19.74 | Indian | Uk37 | Lauterbach et al. (2020) |
| 29 | GIK17940_2 | 117.4 | 20.1 | Pacific | Uk37 | Pelejero et al. (1999) |
| 30 | GiK18515_3 | 119.4 | -3.6 | Pacific | MgCa (G. ruber white) | Schröder et al. (2016) |



| ID | Core name | Lon [°E] | Lat [°N] | Ocean basin | Sensor | Reference |
|----|-----------|----------|----------|-------------|--------|-----------|
| 31 | GIK18519_2 | 118.1 | -0.6 | Pacific | MgCa (G. ruber white) | Schröder et al. (2018) |
| 32 | GIK18522_3 | 119.1 | 1.4 | Pacific | MgCa (G. ruber white) | Schröder et al. (2018) |
| 33 | GIK18526_3 | 118.2 | -3.6 | Pacific | MgCa (G. ruber white) | Schröder et al. (2018) |
| 34 | GIK18540_3 | 119.6 | -6.9 | Pacific | MgCa (G. ruber white) | Schröder et al. (2018) |
| 35 | GIK23415_9 | -19.1 | 53.1 | Atlantic | Plankt. foram. assembl. | Weinelt et al. (2003) |
| 36 | GL1090 | -42.5 | -24.9 | Atlantic | MgCa (G. ruber white) | Santos et al. (2017) |
| 37 | H214 | 177.4 | -36.9 | Pacific | Plankt. foram. assembl. | Samson et al. (2005) |
| 38 | JR244_GC528 | -58.0 | -53.0 | Atlantic | Uk37 | Roberts et al. (2016, 2017) |
| 39 | KNR159_5_36 | -46.5 | -27.5 | Atlantic | MgCa (G. ruber white) | Carlson et al. (2008) |
| 40 | LV29_114_3 | 152.9 | 49.4 | Pacific | MgCa (N. pachyderma) | Riethdorf et al. (2013) |
| 41 | M35003_4 | -61.2 | 12.1 | Atlantic | Uk37 | Rühlemann et al. (1999) |
| 42 | M35003_4 | -61.2 | 12.1 | Atlantic | Plankt. foram. assembl. | Hüls and Zahn (2000) |
| 43 | M77_2_059_1 | -81.3 | -4.0 | Pacific | MgCa (G. ruber white) | Nürnberg et al. (2015) |
| 44 | M77_2_059_1 | -81.3 | -4.0 | Pacific | MgCa (N. dutertrei) | Nürnberg et al. (2015) |
| 45 | M77_2_059_1 | -81.3 | -4.0 | Pacific | Uk37 | Nürnberg et al. (2015) |
| 46 | MD01_2378 | 121.8 | -13.1 | Indian | MgCa (P. obliquiloculata) | Xu et al. (2006, 2008) |
| 47 | MD01_2378 | 121.8 | -13.1 | Indian | MgCa (G. ruber) | Xu et al. (2006, 2008) |
| 48 | MD01_2416 | 167.7 | 51.3 | Pacific | Plankt. foram. assembl. | Gebhardt et al. (2008) |
| 49 | MD01_2416 | 167.7 | 51.3 | Pacific | MgCa (N. pachyderma) | Gray et al. (2018) |
| 50 | MD02_2489 | -148.9 | 54.4 | Pacific | Plankt. foram. assembl. | Gebhardt et al. (2008) |
| 51 | MD02_2575 | -87.1 | 29.0 | Atlantic | MgCa (G. ruber white) | Ziegler et al. (2008) |
| 52 | MD06_3067 | 126.5 | 6.5 | Pacific | MgCa (G. ruber) | Bolliet et al. (2011) |
| 53 | MD06_3067 | 126.5 | 6.5 | Pacific | MgCa (P. obliquiloculata) | Bolliet et al. (2011) |
| 54 | MD88_770 | 96.5 | -46.0 | Indian | Plankt. foram. assembl. | Labeyrie et al. (1996) |
| 55 | MD95_2039 | -10.3 | 40.6 | Atlantic | Plankt. foram. assembl. | Salgueiro et al. (2014) |
| 56 | MD95_2042 | -10.2 | 37.8 | Atlantic | Uk37 | Pailler and Bard (2002) |
| 57 | MD95_2043 | -2.6 | 36.1 | Atlantic | Uk37 | Cacho et al. (1999) |
| 58 | MD98_2181 | 125.8 | 6.3 | Pacific | MgCa (G. ruber) | Stott et al. (2002, 2007) |
| 59 | MD98_2181 | 125.8 | 6.3 | Pacific | MgCa (T. sacculifer) | Stott et al. (2002) |
| 60 | NA87_22 | -14.6 | 55.5 | Atlantic | Plankt. foram. assembl. | Vogelsang et al. (2001) |
| 61 | PS75_056_1 | -114.8 | -55.2 | Pacific | diatom assemblages | Benz et al. (2016) |
| 62 | RAPiD_15_4P | -17.1 | 62.3 | Atlantic | MgCa (N. pachyderma) | Thornalley et al. (2011) |
| 63 | RS147_GC07 | 146.3 | -45.2 | Indian | Uk37 | Sikes et al. (2009) |





| ID | Core name | Lon [°E] | Lat [°N] | Ocean basin | Sensor | Reference |
|----|-----------|----------|----------|-------------|--------|-----------|
| 64 | RS147_GC07 | 146.3 | -45.2 | Indian | Plankt. foram. assembl. | Sikes et al. (2009) |
| 65 | SO201_2_12KL | 162.4 | 54.0 | Pacific | MgCa (N. pachyderma) | Riethdorf et al. (2013) |
| 66 | SO201_2_85 | 170.4 | 57.5 | Pacific | MgCa (N. pachyderma) | Riethdorf et al. (2013) |
| 67 | SO42_74KL | 57.3 | 14.3 | Indian | Plankt. foram. assembl. | Schulz (1995) |
| 68 | SU81_18 | -10.2 | 37.8 | Atlantic | Uk37 | Bard et al. (2000) |
| 69 | SU81_18 | -10.2 | 37.8 | Atlantic | Plankt. foram. assembl. | Vogelsang et al. (2001) |
| 70 | TR163_22 | -92.4 | 0.5 | Pacific | MgCa (G. ruber) | Lea et al. (2006) |
| 71 | V25_59 | -33.5 | 1.4 | Atlantic | Plankt. foram. assembl. | Waelbroeck et al. (1998) |
| 72 | WIND_28K | 51.0 | -10.2 | Indian | MgCa (G. ruber white) | Kiefer et al. (2006) Johnstone et al. (2014) |
| 73 | WIND_28K | 51.0 | -10.2 | Indian | MgCa (T. sacculifer) | Johnstone et al. (2014) |
| 74 | WIND_28K | 51.0 | -10.2 | Indian | MgCa (N. dutertrei) | Kiefer et al. (2006) Johnstone et al. (2014) |



**Table 3.** Characteristics of the example PPE and the two sets of PPEs described in Sect. 3.3. For set 1, all combinations of reference simulations, pseudo-proxy SNRs, and pseudo-proxy noise types are employed with the same settings for pseudo-proxies and forward-modeled proxy time series. For set 2, the 12 combinations of reference simulations, pseudo-proxy SNRs, and pseudo-proxy noise types are employed with all combinations of forward-modeled proxy time series SNRs and noise types.

| Name | Reference simulations | Pseudo-proxy SNRs | Pseudo-proxy noise types | Forward-modeled proxy SNRs | Forward-modeled proxy noise type |
|---|---|---|---|---|---|
| Example | MPI_Glac1D_P3 | 1.6 | AR1 (1289 yrs) | As pseudo-proxies | As pseudo-proxies |
| Set 1 | All ensemble members | 1/4, 1/2, 1, 2, 4, 8, 16, Inf | White AR1 (1000 yrs) power-law | As pseudo-proxies | As pseudo-proxies |
| Set 2 | All ensemble members | 2 | AR1 (1000 yrs) | 1/4, 1/2, 1, 2, 4, 8, 16, Inf | White, AR1 (1000 yrs), power-law |