# Peer review of "Towards spatio-temporal comparison of simulated and reconstructed sea surface temperatures for the last deglaciation"

_EGUsphere, 2023_

## Author Response (AR1)

**Response letter**

Nils Weitzel, Heather Andres, Jean-Philippe Baudouin, Marie-Luise Kapsch,
Uwe Mikolajewicz, Lukas Jonkers, Oliver Bothe, Elisa Ziegler,
Thomas Kleinen, André Paul, and Kira Rehfeld

We thank the Reviewers and the Editor for assessing our manuscript and providing constructive feedback that strengthens our manuscript. The main changes to the manuscript are as follows:

- We incorporated a clearer separation between the seven simulations with transiently changing orbital, GHG, and ice sheet forcing (TraCE-ALL and the six MPI-ESM simulations) and three sensitivity experiments (TraCE-GHG, TraCE-ORB, FAMOUS) which change either only one boundary condition or apply changes in an accelerated manner. To this purpose, we modified Fig. 1, 7, 8, and 9, restructured Sect. 4.2, and adapted Sect. 5.2.

- We reworked Sect. 5.2 to integrate additional analysis (rank scores) assessing statistical differences of the model-proxy agreement of the six MPI-ESM simulations and TraCE-ALL, improve the clarity of the text, and include suggestions by the reviewers.

- We merged the Sect. 4.1.2 and 4.1.3 and moved the previous Fig. 6 to the supplemental information to shorten the technical part of the manuscript.

- We added a new Fig. 6 visualizing the regional temperature evolution of simulations and reconstructions. This figure provides a first impression of the model-proxy agreement and can support the understanding of the quantitative model-data comparison results.

- We made several minor changes to the text to improve its clarity and incorporate suggestions by the Reviewers.

The following is a point-by-point response to the comments by the Reviewers. Here, the comments by the Reviewers are shown italicized in teal and our responses are provided in black.

**Response to Reviewer 1**

*While I am not an expert in the statistical models used within the methodology, the body of work appears sound without any immediate issues.*

We thank the Reviewer for this assessment of our work.

*I feel the wording justifying the need for the extra complexity of using a PSM could be improved. One of the core justifications for using the PSM is it avoids the need to rely on "sparse and uncertain proxy data", yet the PSM largely degrades or reforms the model output to be more compatible with the proxy data and then uses same sparse and uncertain proxy data to benchmark the PSM created forward-modeled proxy time series. The authors also do not address why more traditional signal processing methods, such as Principal Component Analysis, could not instead be used to extract signal from the messy proxy data without the added complexity (and caveats) of using the PSM. I don't think these criticisms undercut the work in any way, only that the justification for the need of the algorithm and PSM could be framed better.*

We thank the Reviewer for the suggestions. We do already discuss advantages of using PSMs in a forward-modeling approach in different parts of the manuscript. We summarize all of the advantages in the Introduction of the revised manuscript. With respect to the quote included in the comment of the Reviewer, we note that the full quote about *"sparse and uncertain proxy data"* is that *"The use of proxy forward modeling avoids the need to reconstruct gridded or regional mean temperatures from sparse and uncertain proxy data"*. This statement was supposed to say that proxy forward modeling avoids the reconstruction of gridded fields or regional mean time series, not that it avoids using sparse and uncertain proxy data. Of course, any model-data comparison method using paleoclimate proxies has to handle the sparse and uncertain nature of this type of data. In the revised manuscript, we rephrase the statement at hand as follows to avoid misunderstandings: *"The use of proxy forward modeling [...] avoids the need to reconstruct gridded fields or regional mean temperature time series from sparse and uncertain proxy data"*.

The main advantages of using PSMs are that, by accounting for non-climatic processes, (1) they allow to correct for potential biases in (timescale-dependent) temperature variability estimates from proxy data (e.g., Laepple and Huybers, 2014) and (2) they can help to distinguish patterns of temperature variations from non-climatic variations in the proxy time series. Using PSMs in a forward-modeling approach, as the algorithm described in Sect. 3.1 does, avoids the need to interpolate reconstructed temperature time series to a common time axis or spatial grid from the sparse and non-uniformly distributed records. While PSMs can be inverted using Bayesian statistics to reconstruct temperature time series with a regular time axis, this inversion is computationally expensive and requires the definition of prior distributions for the temperature time series which are hard to estimate from proxy data. Therefore, fewer assumptions and computational resources are needed in the forward approach. In the revised manuscript, we add the incorporation of the proxy understanding as an advantage of using PSMs in the abstract and conclusions as follows: *"The use of proxy forward modeling allows accounting for non-climatic processes, that affect the temperature reconstructions. [...]"*.

We have not addressed problems with more established signal processing methods because, to our knowledge, none of them has been used previously for quantitative spatio-temporal comparisons of proxies and simulations for the last deglaciation. This would have led to an arbitrary choice of methods to compare our methodology to. Most traditional signal extraction methods, in particular principal component analysis, are not designed to extract signals from irregularly spaced time series with varying time axes and coverage periods. To apply a PCA, all the data needs to be interpolated to the same time axis, which requires similar temporal resolutions of the proxy records and a common coverage interval. Additionally, (1) applying PCAs to reconstructions and simulations still requires the development of a comparison metric between the resulting spatial pattern / time series, which needs to account for uncertainties in the PCAs, (2) PCAs have stricter assumptions on the distributions of the non-climatic noise than PSMs which can lead to biases if these are violated, and (3) results of a PCA depend on an a priori defined region of interest while it is possible in our method to a posteriori assess arbitrary regions. Due to the absence of previous examples that employ standard signal processing methods for spatio-temporal model-data comparison for the last deglaciation, we prefer to focus on better explaining the advantages of using proxy-forward modeling in the revised manuscript instead of discussing why more standard signal processing methods are less suited for this task.

*While less text is dedicated to evaluating the model simulations against data, I disagree with one of*

[Figure]

Figure 1: Rank scores for the ensemble of the six MPI-ESM simulations and TraCE-ALL. Rankings are computed for each proxy record and each of the four components. The bars depict the occurrence percentages of the ranks.

*the key findings: "Comparing the MPI-ESM and CCSM3 simulations that employ orbital, GHG, and ice sheet forcing, we find no systematic differences between the two climate models. In particular, TraCE-ALL is mostly within the IQD spread of the six MPI-ESM simulations." This is an important statement, so the language should be more precises. What specifically are the authors referring to? There is only one MPI-ESM simulation that employs orbital, GHG, and ice sheet forcing (MPI_Ice6G_P2_glob), and there is no comparable TraCE simulation since TraCE-GHG and TraCE-ORB fix the ice sheet forcing to LGM (see Table 1). When I look at Figure 7, I don't see TraCE-ALL effectively being the same as MPI with freshwater flux. Especially in the North Atlantic and North Pacific. [...] The TraCE-ALL and hosed MPI IQDs in the Figure 9 legend are largely not similar, no less TraCE-ALL bracketed by MPI. It is often difficult and nuanced to say when one model is performing better than another, but I don't think the analysis and figures here support the claim TraCE-ALL and hosed MPI are effectively the same when compared to data.*

We apologize for the imprecise language in the paragraph mentioned by the Reviewer. The statement aimed at a comparison of the IQDs of TraCE-ALL with the IQDs of the six MPI-ESM simulations (which all employ orbital, GHG, and ice sheet forcing, but inject freshwater into the ocean in different ways) across regions and components of the deglacial temperature evolution. We did not want to make the statement that TraCE-ALL and the MPI-ESM simulations are indistinguishable. Instead, we wanted to say that there is currently no clear evidence to identify differences in the ability of the employed models, MPI-ESM and CCSM3, to simulate the last deglaciation, because the experiment protocols (in particular regarding the location, timing, and magnitude of freshwater injections) are too different to attribute results clearly to either model differences or differences in the experiment protocol. We reformulate Sect. 5.2 to improve the messaging.

To compare the IQDs of TraCE-ALL and the six MPI-ESM simulations across regions and components of the deglacial temperature evolution in a more quantitative way, we now compute rank statistics that we use in the revised manuscript (see Fig. 1). These are computed for a reduced ensemble, which excludes TraCE-ORB, TraCE-GHG and FAMOUS. Rankings are computed for each proxy record and each of the four components. The average (over all records and components) of the simulations is between 3.8 (for MPI_Glac1D_PTK and MPI_Ice6G_P2_glob) and 4.2 (for MPI_Glac1D_P3) with TraCE-ALL having an average rank of 4.0. This shows that we cannot find a systematically superior performance for any of the seven simulations across regions and components, despite the different experimental protocols and employed models. Meanwhile we find a difference when comparing the rank statistics in more detail. The ranks of TraCE-ALL are concentrated at 1 (highest agreement) and 7 (lowest agreement) whereas the MPI-ESM simulations have much flatter rank histograms. This suggests that TraCE-ALL is more often outside than inside the range of the MPI-ESM simulations even though it does not feature a systematically higher or lower rank. This is in contrast to our previous statement of *"TraCE-ALL is mostly within the IQD spread of the six MPI-ESM simulations"* which we removed from the manuscript. This concentration of TraCE-ALL at extreme ranks tends to hold for all four components (see Fig. 2 - Fig. 5). At the moment, we cannot determine whether these differences originate from the varying experiment protocols or model formulations. We include the rank statistics in the supplemental information of the revised manuscript and refer to them in the revised Sect. 5.2.

[Figure]

Figure 2: As Fig. 1, but restricted to orbital magnitude IQDs.

[Figure]

Figure 3: As Fig. 1, but restricted to orbital pattern IQDs.

[Figure]

Figure 4: As Fig. 1, but restricted to millennial magnitude IQDs.

[Figure]

Figure 5: As Fig. 1, but restricted to millennial pattern IQDs.

*What does it mean that MPI_Ice6G_P2_noMW outperforms TraCE-ALL in the North Atlantic for orbital pattern, when TraCE-ALL is specifically designed to reproduce the reconstructed AMOC variability? AMOC variability is sub-orbital scale of course, but what does it mean that a model without hosing is capturing that scale of variability better than TraCE-ALL (or conversely, the addition of hosing degrades the orbital-scale performance)? Likewise, what are the implications of TraCE-ALL and TraCE-GHG having nearly identical millennial pattern deviations in the North Atlantic, when the TraCE-GHG doesn't include freshwater hosing?*

As we explain in Sect. 4.2 and 5.2, these results originate from regional differences between the Mediterranean North Atlantic and the Subpolar North Atlantic in the reconstructions, which are not present to the same degree in the simulations. TraCE-ALL agrees the most with the reconstructions in the Mediterranean North Atlantic but its performance in the whole North Atlantic is degraded by poor agreement in the Subpolar North Atlantic. When combining magnitude and pattern metrics in biplots (see Fig. 7 and response to comment below), simulations with local freshwater injection perform the best in the North Atlantic for either timescale, MPI_Glac1D_P3 for orbital timescales and TraCE-ALL for millennial timescales.

When focusing purely on the patterns, the results mentioned by the Reviewer have several potential implications. First, a good agreement for orbital timescales does not imply a good agreement for millennial timescales and vice versa. This could be due to varying importance of forcings and internal feedback processes for the patterns of temperature changes on different temporal and spatial scales. Second, reproducing the patterns of a small set of proxies might be an insufficient strategy to capture the spatial structure of millennial-scale temperature patterns. For example, our results suggest that reproducing the patterns of a specific AMOC proxy (e.g., Pa/Th ratios at Bermuda rise), as TraCE-ALL does, will not necessarily lead to a good model-proxy agreement for millennial-scale temperature patterns across different regions. Instead, other factors such as the magnitude of the AMOC response, which cannot be directly quantified from comparing Pa/Th ratios with simulated AMOC strengths, or the background climatic state could potentially have a large influence on the regional manifestations of the temperature variability patterns. Third, the uncertainty about the factors controlling millennial-scale variability could lead to an adequate reproduction of the pattern of AMOC variability with an incorrect mechanism. In this case, the spatially varying degree of model-proxy agreement could be the result of an incorrect driver of millenial-scale variability. The uncertainties of ice sheet and freshwater flux reconstructions, and the heterogeneity of the experiment protocols make it very challenging to determine the reasons for the spatially varying performance. Addressing these challenges with designated protocols in the context of inter-model comparison projects could be a promising way forward. In the revised manuscript, we enhance the discussion of potential implications of the varying model-proxy agreements on different temporal and spatial scales, and give more suggestions for future research to address the identified challenges.

*The text notes "More generally, all simulations with meltwater input show a better agreement with reconstructions for millennial magnitudes than those without meltwater input." I don't think this is strictly true. There are cases where MPI_Ice6G_P2_noMW performs similar to, if not better than, the routed MPI-ESM simulations. In either case, this only means the millennial-scale variability is more like the data when hosing is added, not that the pattern is realistic (as noted around line 550). This is more apparent with the TraCE simulations where in some locations the addition of hosing degrades model performance. Getting the magnitude of variability correct, but the patterns (ie trends) of the deglaciation wrong isn't particularly satisfying, which could be emphasized here.*

We apologize for the imprecise statement in l. 546 of the original manuscript. We aimed at referring strictly to the global averages in this statement. Indeed, MPI_Ice6G_P2_noMW performs similar or better than the simulations with local freshwater input in some regions. In the revised manuscript, we replace the sentence starting with *"More generally..."* by *"In particular, in the global average, all simulations with meltwater input show a better agreement with reconstructions for millennial magnitudes than those without meltwater input."*

We agree with the Reviewer's statement that this only implies that *"millennial-scale variability is more like the data when hosing is added, not that the pattern is realistic"*. As we state in l. 552ff, we can currently not determine the reason for the absence of pattern improvement when including local freshwater injections. Following the advice of the Reviewer, we add the following sentence in the same paragraph to emphasize this contrast between magnitudes and patterns more strongly: *"The contrast between higher model-proxy-agreement in simulating millennial magnitudes*

*but no improvement for millennial patterns in the fully forced simulations hints at limitations in our current understanding of the spatio-temporal structure of millennial-scale variability during the deglaciation."*

*Line 125: define or give examples of "sensor" for the novice.*

Following this suggestion, we define sensors and proxies as following in the revised manuscript: *"Here and in the following, we refer to sensors as the organisms recording the temperature signal (e.g., planktonic foraminifera) and proxies as the measured temperature-sensitive quantities (e.g., Mg/Ca ratios, species compositions)."* This definition is given in the Introduction, when sensors and proxies are first mentioned.

*Line 137: Osman, et al., 2021 uses four proxy types, so I am not sure why it cited here.*

We apologize for the imprecise language in the sentence. Both cited studies use multiple, but distinct sensors (Paul et al. (2021) uses only faunal and floral assemblages whereas Osman et al. (2021) uses only geochemical proxies, i.e., Mg/Ca, $U_{37}^k$, $TEX_{86}$ and $\delta^{18}O_c$). We included the half sentence to emphasize that we use both, assemblage-based and geochemical proxies. In the revised manuscript, we correct the sentence as follows: *"Unlike some recent studies focusing on either assemblage-based temperature reconstructions (e.g., Paul et al., 2021) or geochemichal proxies (e.g., Osman et al., 2021), we employ a multi-proxy approach using the calibrations proposed by the original authors for assemblages and geochemical proxies, respectively.".*

*Lines 177: "Computing averages in this last step instead of averaging temperature time series in the beginning avoids interpolating proxy records with irregular time axes to a common resolution." Explain this. It seems like in some portions of the analysis the data are binned to a fixed 100-yr timestep (ie Section 3.2). The time series displayed in Figure 9 are regional averages, are these first binned to 100-yr interval or are they somehow calculated on irregular time spacing for multiple records and ensemble members?*

The Reviewer is correct that the used data is interpolated to a regular time axis with 100 yr time steps in Sect. 3.2 (Estimation of proxy system model parameters) and to construct the regional stacks in Fig. 9. We want to note that we consider neither the estimation of the proxy system parameters nor the construction of the regional mean stacks as part of the model-data comparison algorithm. Only the four steps described in Sect. 3.1 are part of the algorithm. Therefore the statement in line 177 refers just to the procedure described in those four steps and motivates the averaging of IQDs obtained for individual records instead of averaging temperature records before computing the deviations between reconstructions and simulations.

The interpolation of the reconstructions to a common time axis in the estimation of the PSM is suboptimal. As the core topic of the paper is the development and testing of the new algorithm, additionally developing a more sophisticated method, which avoids interpolation, to estimate the PSM parameters is beyond the scope of our manuscript. We mention this limitation explicitly in the discussion of the revised manuscript and recommend the development of more advanced methods to estimate the temporal structure of the non-climatic noise process in addition to the integration of process-based PSMs by adding the following two sentences in Sect. 5.2: *"Moreover, our procedure to estimate the PSM parameters requires interpolating the proxy records to a common time axis which is otherwise avoided in the model-data comparison algorithm. Developing a more sophisticated method for the parameter estimation would be beneficial for future applications of our algorithm."*

Fig. 9 only serves illustrative purposes to aid the interpretation of the quantitative model-data comparison results. The IQD values in the legends of the respective panels are the averaged IQDs of the records included in the construction of the stacks, but they are not computed by comparing the reconstructed and simulated stacks quantitatively. We are sorry for not explicitly noting this in the previous manuscript version and adjust the caption of Fig. 9 to emphasize the role of the regional stacks by adding the following two sentences: *"Note that the stacks are not used in the model-data comparison algorithm, but just provide a visual impression of the reconstructed and simulated regional temporal evolution. The methodology to construct the stacks is described in the supplemental information (Text S5)."*

*Lines 220: magnitude is defined as the standard deviation of each ensemble member (for the decomposed time series). Since standard deviation is an absolute value, doesn't this fail to discriminate between trends in opposite directions (ie data is cooling when model is warming)?*

Indeed, the magnitude metric does not discriminate between trends in opposite directions but only measures the strength of variations. However, opposing trends lead to very high pattern IQDs (e.g., orbital pattern IQDs for TraCE-ORB in the Southern Hemisphere) as opposing trends create large deviations of the 'pattern' time series. To evaluate strength and direction of trends together, one needs to jointly consider magnitude and pattern IQDs. As the Reviewer suggests below this can be visualized by using biplots of the two metrics in which simulations that match the reconstructed strength and direction of trends appear in the lower left (see Fig. 7). We note that trends during the deglaciation are mostly non-linear. Therefore, reducing the orbital signal purely to the strength and direction of trends does not seem sensible. This motivates our separation into magnitudes and patterns. In the revised manuscript, we try to better explain the separation into magnitudes and patterns in Sect. 3.1. Additionally, we emphasize that a single metric is likely insufficient for fully capturing the deviations between simulations and reconstructions in an interpretable way. The use of biplots as suggested by the Reviewer below is mentioned explicitly as a way to jointly assess magnitudes and directions (or, more generally, patterns) of trends.

*Throughout the text "magnitude" is used to denote the degree of variability in the decomposed time series, which is just saying the strength of variability. It may not be obvious to the reader what the utility of this metric is. We tend to think in terms of time series, so "pattern" (as defined here) is far more intuitive.*

Indeed, we use the term 'magnitude' to denote the strength of variability. We do somewhat disagree with the notion that patterns are *"far more intuitive"* than magnitudes. Patterns are mostly meaningful if variations are externally forced and if there are sufficiently tight constraints on the boundary condition reconstructions such that models can be expected to reproduce the observed patterns of variation. As we discuss in l. 583-589 of the original manuscript, whether these two conditions are fulfilled, is debated. Models forced with local freshwater injections computed from established ice sheet reconstructions do not reproduce the observed millennial-scale patterns in the North Atlantic (Kapsch et al., 2022) while another model (MIROC) produced a spontaneous abrupt warming similar to the transition into the Bolling-Allerod (Obase and Abe-Ouchi, 2019). Therefore, we think that comparing patterns of time series alone is an insufficient metric to comprehensively evaluate simulations of the last deglaciation. Instead, we attempt to combine this pattern approach with methods developed for analysing unforced variability in simulations and proxies for e.g. the LGM, the Holocene, and the last millennium (e.g., Laepple and Huybers, 2014; PAGES 2k Consortium, 2019; Rehfeld et al., 2018). In addition, comparing magnitudes of variations is important for assessing the strength of feedback processes, which in turn can be relevant to constrain the projected response to future emission scenarios or the impact of potential tipping points in the Earth system. We try to better motivate the separation into magnitudes and patterns in Sect. 3.1 of the revised manuscript by adding the following sentences:

*"Magnitude components quantify the strength of timescale-dependent variations, independent of their specific timing. Therefore, they are valuable for assessing the strength of the response to forcing, of spontaneous fluctuations, and of variations forced by time-uncertain boundary conditions. In contrast, pattern components assess the direction, timing, and succession of timescale-dependent variations. They are particularly meaningful if variations are externally forced and if there are sufficiently tight constraints on the boundary condition reconstructions such that models can be expected to reproduce the timing of the observed pattern of variations."*

*Lines 235: Since N is either 100 or 1000, I assume an empirical probability distribution is used rather than a fitted distribution.*

Yes, we are using empirical distributions to approximate the analytically intractable distributions $\mathbb{P}$ and $\mathbb{Q}$. We note this in l. 244 of the original manuscript, when stating *"We compute the IQD using a MC approximation of Equ. (2) with the MC samples from step 2."*. We apologize that this sentence alone has not clarified that we use empirical distributions. In the revised manuscript, we rearrange the paragraphs such that the Monte Carlo approximation is mentioned immediately after explaining the different terms in Equ. (2). In addition, we add the following sentence to state that we approximate $\mathbb{P}$ and $\mathbb{Q}$ by empirical distributions: *"Thereby, we approximate the analytically intractable distributions $\mathbb{P}$ and $\mathbb{Q}$ by empirical distributions."*

*Line 244: How does IQD integrate differences in time series? Each forward-modeled proxy time series is on the same irregular age model spacing as the proxy data, but how is the time series translated to distributions used in the IQD equation?*

In the Monte Carlo approach, we obtain N pairs of realizations of the reconstructed and forward-modeled proxy time series. Each realization has M samples and each of the M samples corresponds

to one depth in the proxy record. Thus, the realizations can be interpreted as an empirical, M-dimensional probability distribution with correlations between the samples due to auto-correlation in the time series. The IQD computes the difference between these M-dimensional distributions by using a Monte Carlo approximation of Equ. (2), which we describe in more detail in the supplemental information. In the revised manuscript, we try to better describe this translation between time series and multivariate probability distributions in Sect. 3.1.2 and 3.1.3.

*Lines 254: Zonal IQD seems to only be used in part of Figure 2, which is a flowchart of the analysis. If it is not used in the results section, could it be removed?*

As the Reviewer correctly points out, we do not use zonal IQDs in Sect. 4.2. This is due to differences between the North Atlantic and North Pacific detected in an initial visual inspection of the reconstructions and simulations. However, we do use zonal mean IQDs in the pseudo-proxy experiments of Sect. 4.1. The evaluation of the pseudo-proxy experiments is meant to assess the general properties of the algorithm, whereas the regions chosen in Sect. 4.2 are more targeted towards the specific data at hand. As we show in Sect. 4.1, the spatial aggregation is an important step to improve the reliability and robustness of the results, independent of the specific partitioning into regions. Therefore, we think that the visualization of this step should not be removed in Fig. 2. As zonal averages are an easy to understand and often-used separation into regions, we decided to use these in Fig. 2 and Sect. 4.1. and do not think that a change to different regions in Fig. 2 is needed to improve the understanding of the steps of the algorithm. In the revised manuscript, we mention that this is just an exemplary partition into regions in the caption of Fig. 2 as follows: *"As an exemplary partition into regions, we show zonal mean IQDs in the bottom row for all latitudinal bands containing at least five proxy records (see Sect. 3.1.4 for a definition of the zonal mean averaging procedure)."*

*Section 4.2 Comparison of simulations against SST reconstructions: I think it would be really useful to the reader to plot orbital + millennial time series for the regions summarized in Figure 7 (ie this new plot should come before Figure 7). I envision something like Figure 9, which would give the reader a feel for what the models are simulating (relative to the data) prior to the decomposition. Those raw trends are somewhat abstracted away by plotting orbital and millennial time series separately. For example, in Figure 9 MPI_Ice6G_P3 has a cooling trend around 14 – 13 ka in both the orbital and millennial scales. If it is caused by the injection of freshwater forcing, I would expect it to only be in the millennial-scale (also perhaps implying freshwater forcing is showing up in the orbital-scale decomposition).*

The Reviewer suggests an additional visualization of reconstructions and simulations prior to the quantitative model-data comparison. We did not include a visualization as suggested by the reviewer in the original manuscript as the focus of the manuscript is the method development and not an all-encompassing assessment of the model-proxy agreement. Nevertheless, we understand that a figure as described by the Reviewer can help the readers to get a better impression of the data that we use. Therefore, we created regional stacks for the four disjunct regions (Southern Hemisphere extratropics, Tropics, extratropical North Atlantic, extratropical North Pacific) of reconstructed and forward-modeled temperature time series. The procedure is similar to Fig. 9, but for the four disjunct regions from Fig. 7 and before employing the timescale decomposition and the decomposition into magnitudes and patterns of variations (Fig. 6). We include this figure as the new Fig. 6 in the revised manuscript and move the previous Fig. 6 to the supplemental information (see response to comments of Reviewer 2). As noted by the Reviewer, strong freshwater-water induced perturbations can have an imprint on the orbital-scale signal, albeit weaker than in the millennial-scale signal. This is the case when the perturbations are large enough to substantially influence time-averages on timescales longer than the smoothing frequency used to separate orbital-scale and millennial-scale variations. The cooling trend around 14 ka in MPI_Ice6G_P3 is one example of this phenomenon. We add a sentence in Sect. 5.2 of the revised manuscript to discuss this interaction between orbital- and millennial timescales.

*Figure 7: It would be very verbose, but would it be worth plotting a magnitude versus pattern IQD scatter plot? There are too many combinations for the main text, so perhaps an example (perhaps millennial magnitude versus pattern for the North Atlantic)? The best models should converge in the lower left of the plot near the plot origin (0,0).*

We thank the Reviewer for this very insightful visualization idea. In particular since magnitude and pattern scores provide different information on the model-proxy agreement, combining these two aspects in one plot can be a helpful visualization. As the Reviewer correctly notes, the best performing models are located in the lower left corner. We have decided to not include this figure

[Figure]

Figure 6: Regionally stacked SST variations for records in (a) the Southern Hemisphere extratropics (n=10 proxy records), (b) the Tropics (n=44), (c) the extratropical North Atlantic (n=13), and (d) the extratropical North Pacific (n=7). Black lines denote the stacked reconstructions, whereas colored lines depict the stacked forward-modeled proxy time series derived from the ten transient simulations. Shaded areas show uncertainties from chronologies and the PSM. Note that the stacks are not used in the model-data comparison algorithm, but just provide a visual impression of the reconstructed and simulated regional temporal evolution. The methodology to construct the stacks be described in the supplemental information of the revised manuscript.

in the main text as it does not add a new information compared to Fig. 7. However, we include it as supplemental figure for globally and regionally averaged IQDs and reference it in Sect. 4.2. We include one version with all ten simulations and one version without the three 'sensitivity experiments', which either change just one boundary condition transiently (TraCE-ORB, TraCE-GHG) or apply boundary condition changes with a 10x acceleration (FAMOUS) (see response to Reviewer 2 for more information on this separation in the revised manuscript) (Fig. 7 and Fig. 8). The latter plot is meant to improve the visualization of differences between the remaining seven simulations in cases where the performance of at least one of the sensitivity experiments differs strongly from the six MPI-ESM simulations and TraCE-ALL (e.g., TraCE-ORB for orbital variations in the Tropics).

*Line 597: "To avoid the need to reconstruct gridded or regional mean temperatures from sparse and uncertain proxy data, the algorithm applies proxy system models to simulation output and quantifies the deviation between the resulting forward-modeled proxy time series and temperature reconstructions". Doesn't Figure 9 create regional stacks? I understand mean IQD is used to summarize regions (as explained in section 3.1.4), but how are time series of regional averages constructed?*

As the Reviewer points out correctly, we construct regional stacks in Fig. 9. As mentioned above, these are not part of the quantitative model-data comparison but are only aiding the interpretation of the regional mean IQDs. The regional mean IQDs in Fig. 7 and in the labels of Fig. 9 are averaged IQDs from each proxy record located in the respective regions. We mention this as *"The numbers in the legends next to each simulation are the averaged IQDs over all records in the respective stacks"*. As using the word *'stacks'* in this statement might have led to confusion, we change it to *'regions'* in the revised manuscript. As we point out in Sect. 3.1.4, averaging as the last step of the model-data comparison avoids the creation of regional mean time series / gridded reconstructions, since we only have to average numbers instead of time series. To formulate this point more clearly, we replace *"reconstruct gridded or regional mean temperatures"* by *"reconstruct gridded fields or regional mean temperature time series"* in the revised manuscript. We describe the construction of the regional stack in the supplemental information of the revised manuscript.

[Figure]

Figure 7: Biplots of IQDs for orbital-scale and millennial-scale variations. The magnitude IQDs of variations are plotted on the x-axes and the pattern IQDs on the y-axes. Lines indicate uncertainties from varying the PSM parameters. Dots in the lower left corner indicate simulations with the highest model-proxy agreement for magnitudes and patterns.

[Figure]

Figure 8: As Fig. 7 but without the three sensitivity experiments TraCE-ORB, TraCE-GHG, and FAMOUS.

*Figure S2. Many of the plot titles in Figure S2 are identical. I assume this is depicting multiple records from the same core site. Perhaps this could be denoted better in the plot titles or figure caption. Also, how are the regional stack time series in Figure 9 made when not all records in Figure S2 span 19 − 9 ka?*

We thank the Reviewer for this observation. The assumption is correct that the identical plot titles occurred in the case of records belonging to the same core. In the revised Fig. S2, we include the IDs of each record from Table 2 in the plot titles, which identify each record uniquely from the combination of core name and sensor.

For the regional stacks, we are considering records only for the periods that they cover, which can differ depending on the selected age ensemble member. This can lead to artefacts, mostly at the edges of the period 19-9 ka. To reduce these artefacts, we restrict the coverage period of the stacks to 19-9 ka whereas the model-data comparison algorithm includes all samples in the interval 22-6 ka. This is not a major issue because the stacks are not used to compute the IQDs but just support the understanding of the model-data comparison results. We apologize for not describing the methodology of computing the stacks before, because they are not part of the model-data comparison methodology. In the revised manuscript, we note explicitly in the figure caption that the stacks are not used to compute the IQDs and describe their construction methodology in the supplemental information.

**Response to Reviewer 2**

*In this manuscript the authors present a new methodology to compare simulated and reconstructed sea surface temperatures and apply it to the case of the last deglaciation. The method nicely separates different aspects of temperature variability on different time-scales and will become a valuable tool to quantify model-data agreement as new transient model simulations and more proxy records of past climate intervals become available. The paper is well written and the results are clearly presented and I therefore recommend publication of the paper in Climate of the Past after some, mostly minor, issues have been addressed.*

We thank the Reviewer for this positive assessment of our work.

*I'm not convinced that the title properly reflects the content of the paper, namely a comparison of simulated and reconstructed sea surface temperatures. Possibly reformulate to something like:*

*Towards spatio-temporal comparison of simulated and reconstructed (sea surface) temperatures for the last deglaciation*

We thank the Reviewer for this suggestion to improve the title of the manuscript. In the revised manuscript, we change the title to *"Towards spatio-temporal comparison of simulated and reconstructed sea surface temperatures for the last deglaciation"*.

*I realize that this is a predominantly methodological paper, but the authors could possibly consider to shorten the technical part a bit by moving some details to the supplementary, and focus a bit more on the results in terms of how well different models reproduce different aspects of the temperature evolution over the last deglaciation. For example Fig. 6 seems to add very little information and the corresponding section 4.1.3 seems very extended considering that the important messages are simply that i) the reliability and robustness of the algorithm seem to be very little affected by misspecified SNRs and ii) the effect of under- or overestimating the temporal persistence of non-climatic noise is negligible in our PPEs.*

We understand the wish of the Reviewer to reduce the technical part of the paper a bit. However, we also have to give the interested reader a chance to understand how we obtain the respective results without extensively consulting the supplemental information. Therefore, we would prefer not to shorten the methods section and instead giving the reader the option to skip technical subsections, e.g. Sect. 3.1.1 to 3.1.4., if they are more interested in the results of the application of the methodology. Following the advice of the Reviewer, we merge Sect. 4.1.2 and 4.1.3 and move the previous Fig. 6 to the supplemental information. We still think that the two relatively simple messages mentioned by the Reviewer are of relevance because they can provide some guidance for future proxy system model developments. In addition, they were not a priori (i.e., before running the pseudo-proxy experiments) obvious to us. Therefore, we still include them in the results section and not move this part of the analysis fully into the supplemental information.

*A nice addition to Figures 8 and 9 would be a figure showing also a direct comparison of simulated and reconstructed temperature (anomalies) time series for the different regions. I believe that a simple visual inspection of model and observation time series is still useful to get a first idea about model-data agreement.*

We thank the Reviewer for this suggestion. We agree that a visualization of the difference is still a useful first step in model-data comparison as long as it does not lead to a rushed judgement on the model-proxy agreement. We did not include it in the original manuscript as the focus of the manuscript is the method development and not an all-encompassing assessment of the model-proxy agreement of the ten simulations. Nevertheless, we understand that a figure as described by the reviewer can help the reader get a better impression of the data that we use. Therefore, we created regional stacks for the four disjunct regions (Southern Hemisphere extratropics, Tropics, extratropical North Atlantic, extratropical North Pacific) of reconstructed and simulated temperatures. The construction of the stacks follows the same methodology as for Fig. 9, but without employing the timescale decomposition and the decomposition into magnitudes and patterns of variations (Fig. 6). We include this figure as the new Fig. 6 in the revised manuscript and move the previous Fig. 6 to the supplemental information (see response to comment above).

*I suggest removing the simulations which do not include all forcings, i.e. TraCE-ORB and TraCE-GHG, from the main analysis. For example, in Fig. 1a it is confusing to show simulations which do not include the full forcing as it gives the impression that the spread among models is even larger than it actually already is. I think it is perfectly fine to include those simulations to test the*

[Figure]

Figure 9: Revised version of Fig. 1 in the manuscript. In the revised version, the sensitivity experiments TraCE-ORB, TraCE-GHG, and FAMOUS are visually separated from the other seven simulations and the ensemble spread in (b) is only computed from the six MPI-ESM simulations and TraCE-ALL.

*methodology, but not when it comes to the actual comparison of how well models simulate different aspects of the last deglaciation (e.g. lines 540-545).*

We thank the reviewer for pointing out that the presentation of the simulation ensemble as currently done can lead to misunderstandings. It is not our intention to give the impression that all ten simulations are equally likely representations of the temperature changes during the last deglaciation. Nevertheless, we believe that there is value to also compare sensitivity experiments with proxy-based reconstructions, e.g., to understand for which regions and components the inclusion of all three major forcings (orbital, GHG, ice sheets) improves the model-proxy agreement the most and if the accelerated application of transient boundary conditions reduces the model-proxy agreement significantly. Therefore, we revise Fig. 2, 7, 8, and 9 such that they contain clear visual separations between the seven simulations, which apply orbital, GHG, and ice sheet forcing without acceleration, and the three sensitivity experiments TraCE-ORB (only orbital), TraCE-GHG (only GHG), and FAMOUS (accelerated application of forcings, and neither change of land-sea mask nor Antarctic ice sheet). Examples of the adapted figures are provided in Fig. 9 and 10). Furthermore, we adapt Sect. 2.1, 4.2, and 5.2 to better separate the sensitivity experiments from the other seven simulations. In particular, Sect. 4.2.1 and 4.2.2 only contain descriptions of the model-proxy agreement for the six MPI-ESM simulations and TraCE-ALL. We add a Sect. 4.2.3 that assesses for which regions and components the model-proxy agreement of the three sensitivity experiments deviates the most/least from the other seven simulations.

*Can the presented new methodology, which is here applied to SSTs, in principle also be extended and applied to other variables? It is mentioned that it could be extended to land temperature reconstructions, but what about very different variables like carbon or oxygen isotopes?*

Yes, it is possible to extend our approach to other continuous variables (e.g. carbon or oxygen isotopes), which are simulated by Earth system models or can be linked to simulated variables through PSMs. For these variables, one can either integrate existing proxy systems models into the algorithm (e.g., BAYFOX, Malevich et al., 2019, for the oxygen isotopic composition of planktic foraminifera) or adapt the methodology from our manuscript to estimate the structure of the noise process. Employing the algorithm to proxies with binary or categorical values (e.g. occurrence of sea ice, sediment types) would require more substantial changes to the employed statistical methods. In the revised manuscript, we add the following sentence on the applicability to other continuous variables at the end of Sect. 5.2: *"Finally, it is straightforward to adapt our algorithm for model-proxy comparison of other continuous variables such as oxygen isotopes, in particular if PSMs already exist that link the proxies to one or multiple simulated variables."*

*Some parts of the text are filled with acronyms, which makes it sometimes a bit hard to read. Please consider if the use of acronyms could be reduced, particularly also in figure captions. One example: is LD really needed?*

We are sorry for reducing the readability of the text by using many acronyms. We intended to use well-established acronyms (e.g. LGM, GHG, PSM) plus introducing some additional ones to avoid having to repeat lengthy phrases while maintaining sufficient precision of the wording. We

[Figure]

Figure 10: Revised version of Fig. 7 in the manuscript. In the revised version, the sensitivity experiments TraCE-ORB, TraCE-GHG, and FAMOUS are visually separated from the other seven simulations.

understand that this led to some hard-to-read paragraphs. We carefully went through the list of used acronyms and remove the acronyms LD (last deglaciation) and MC (Monte Carlo) in the revised manuscript. We do not remove the other used acronyms completely but replace them by alternative phrases at appropriate places.

*L. 30: The references cited do not support the 3-8°C range given in the paper. From the abstracts: Tierney: -5.7 to -6.5°C; Annan: -4.5+-0.9°C.*

We thank the Reviewer for correctly pointing out the inconsistency of the given range with the references. In the revised manuscript, we correct the value to 3.6-6.5 K as the combined range of the two cited studies.

*Section 2: consistently use either past or present tense*

We thank the Reviewer for pointing out the inconsistent use of past and present tense in Sect. 2. The intention of mixing past and present tense was to clearly separate our choices (in present tense) and choices made during the construction of the simulations / proxy database. However, (a) this strategy was employed inconsistently by us and (b) we realize that it reduces the readability of the section and can lead to confusion. Therefore, we use present tense consistently in Sect. 2 in the revised manuscript. To emphasize that all of the simulations and proxy records have been published previously, we adapt the wording in the first sentences of Sect. 2.1 and 2.2.

*L. 106: are -> is*

Thanks for finding the typo. We correct it in the revised manuscript.

*L. 130-131: How is that justified? What does sub-surface mean? Is it still in the mixed layer?*

It is well-established that most of the employed proxy records reflect surface or mixed layer conditions, in particular all $U_{37}^k$ and assemblage records, and the majority of Mg/Ca records (e.g., Kucera et al., 2005; Rebotim et al., 2017; Tierney and Tingley, 2018). This justifies our approach. All used sensors occupy a range of depths, which has likely changed in the past, but the environmental and biological controls of the vertical habitat variability are poorly constrained (e.g., Greco et al., 2019; Kretschmer et al., 2018). Therefore, we make the simplifying choice of comparing all records with simulated SSTs. We do not expect that this assumption has a significant effect on our

results. In the revised manuscript, we clarify that most of the proxy records reflect surface or mixed layer temperatures as follows: *"Most of these records reflect surface or mixed layer temperatures (Kucera et al., 2005; Rebotim et al., 2017; Tierney and Tingley, 2018). While the used sensors occupy a range of depths, we denote all samples as sea surface temperature (SST) reconstructions in the following."*

*L. 396: FRPRR -> FPRR ?*

Thanks for finding the typo. We correct it in the revised manuscript.

*Fig. 2 top-right, Proxy System Model: the single line representing the simulation is possibly a bit misleading, as there are 4 time series from different model grid cells that enter the PSM, if I understood correctly.*

We thank the reviewer for pointing out this imprecision. The black line in the top right panel is the result from the interpolation of the simulation data to the proxy location. Thus, it is correct that time series from multiple grid boxes are combined in a weighted average in the black line. However, we believe that plotting the time series of all grid boxes used in the interpolation would not improve the reader's understanding of the algorithm as it would reduce the visual clarity of the figure. Instead, we adapt the legend of this panel to explicitly note that the black line is the simulation interpolated to the proxy site. We believe that this is the best compromise between a detailed explanation of the steps of the algorithm and visual clarity. The legend label in the revised panel read *"Simulation at proxy site"* and we adapt the description of the top right panel as follows: *"We apply a proxy system model (PSM) to the simulated SST fields to first obtain simulated time series interpolated to the proxy locations and then Monte Carlo realizations of forward-modeled proxy time series (top row, right)."*

*Fig. 2 bottom: how are the latitudinal belts defined? By the dashed vertical lines? It is not clear from the figure caption. Please repeat the text in 3.1.4.*

The definition of the latitudinal bands follows the definition in Sect. 3.1.4, i.e., overlapping moving windows of 20° width which move in 5° steps. The dashed gray bars are just plotted in 30° steps to improve the visual orientation of the reader. To not lengthen the caption of the plot further, we do not repeat the text from Sect. 3.1.4 in the caption but instead refer to Sect. 3.1.4 for details on the zonal averaging procedure.

*Fig. 2 Timescale decomposition panels: this is just a detail, but it would be nice and more intuitive if the lines would be plotted and shown in the legends in order of increasing 'smoothing', i.e. 1) Reconstruction, 2) Orbital+millennial, 3) Orbital.*

We thank the reviewer for this suggestion which we incorporate in the revised manuscript.

**References**

Greco, M., Jonkers, L., Kretschmer, K., Bijma, J., and Kucera, M.: Depth habitat of the planktonic foraminifera Neogloboquadrina pachyderma in the northern high latitudes explained by sea-ice and chlorophyll concentrations, Biogeosciences, 16, 3425–3437, https://doi.org/10.5194/bg-16-3425-2019, 2019.

Kapsch, M., Mikolajewicz, U., Ziemen, F., and Schannwell, C.: Ocean Response in Transient Simulations of the Last Deglaciation Dominated by Underlying Ice-Sheet Reconstruction and Method of Meltwater Distribution, Geophys. Res. Lett., 49, https://doi.org/10.1029/2021GL096767, 2022.

Kretschmer, K., Jonkers, L., Kucera, M., and Schulz, M.: Modeling seasonal and vertical habitats of planktonic foraminifera on a global scale, Biogeosciences, 15, 4405–4429, https://doi.org/10.5194/bg-15-4405-2018, 2018.

Kucera, M., Weinelt, M., Kiefer, T., Pflaumann, U., Hayes, A., Weinelt, M., Chen, M.-T., Mix, A. C., Barrows, T. T., Cortijo, E., Duprat, J., Juggins, S., and Waelbroeck, C.: Reconstruction of sea-surface temperatures from assemblages of planktonic foraminifera: multi-technique approach based on geographically constrained calibration data sets and its application to glacial Atlantic and Pacific Oceans, Quaternary Science Reviews, 24, 951–998, https://doi.org/10.1016/j.quascirev.2004.07.014, 2005.

Laepple, T. and Huybers, P.: Ocean surface temperature variability: Large model–data differences at decadal and longer periods, Proc. Natl. Acad. Sci. U.S.A., 111, 16 682–16 687, https://doi.org/10.1073/pnas.1412077111, 2014.

Malevich, S. B., Vetter, L., and Tierney, J. E.: Global Core Top Calibration of $\delta^{18}O$ in Planktic Foraminifera to Sea Surface Temperature, Paleoceanography and Paleoclimatology, 34, 1292–1315, https://doi.org/10.1029/2019PA003576, 2019.

Obase, T. and Abe-Ouchi, A.: Abrupt Bølling-Allerød Warming Simulated under Gradual Forcing of the Last Deglaciation, Geophys. Res. Lett., 46, 11 397–11 405, https://doi.org/10.1029/2019GL084675, 2019.

Osman, M. B., Tierney, J. E., Zhu, J., Tardif, R., Hakim, G. J., King, J., and Poulsen, C. J.: Globally resolved surface temperatures since the Last Glacial Maximum, Nature, 599, 239–244, https://doi.org/10.1038/s41586-021-03984-4, 2021.

PAGES 2k Consortium: Consistent multidecadal variability in global temperature reconstructions and simulations over the Common Era, Nat. Geosci., 12, 643–649, https://doi.org/10.1038/s41561-019-0400-0, 2019.

Paul, A., Mulitza, S., Stein, R., and Werner, M.: A global climatology of the ocean surface during the Last Glacial Maximum mapped on a regular grid (GLOMAP), Clim. Past, 17, 805–824, https://doi.org/10.5194/cp-17-805-2021, 2021.

Rebotim, A., Voelker, A. H. L., Jonkers, L., Waniek, J. J., Meggers, H., Schiebel, R., Fraile, I., Schulz, M., and Kucera, M.: Factors controlling the depth habitat of planktonic foraminifera in the subtropical eastern North Atlantic, Biogeosciences, 14, 827–859, https://doi.org/10.5194/bg-14-827-2017, 2017.

Rehfeld, K., Münch, T., Ho, S. L., and Laepple, T.: Global patterns of declining temperature variability from the Last Glacial Maximum to the Holocene, Nature, 554, 356–359, https://doi.org/10.1038/nature25454, 2018.

Tierney, J. E. and Tingley, M. P.: BAYSPLINE: A New Calibration for the Alkenone Paleothermometer, Paleoceanography and Paleoclimatology, 33, 281–301, https://doi.org/10.1002/2017PA003201, 2018.

---

## Author Response (AR2)

**Response letter**

Nils Weitzel, Heather Andres, Jean-Philippe Baudouin, Marie-Luise Kapsch,
Uwe Mikolajewicz, Lukas Jonkers, Oliver Bothe, Elisa Ziegler,
Thomas Kleinen, André Paul, and Kira Rehfeld

We thank the Editor for assessing our revised manuscript and the constructive feedback to further improve it. As explained in the response below, we have shortened the methods section by moving Sect. 3.1.1 to 3.1.4 to the supplemental information. However, we retained the most important information from these subsections in the main manuscript. The following is a point-by-point response to the comments by the Editor and from the file validation. Here, the comments by the Editor are shown italicized in teal and our responses are provided in black.

*I have now gone through your manuscript and the revisions carefully. I think this is a sound piece of work and a very well written manuscript and that you have correctly addressed most requests of both reviewers.*

We thank the Editor for this assessment of our work.

*My only concern has to do with the length of the manuscript, already mentioned by one of the reviewers. Readers interested in using the method will make the effort to read it carefully but I am afraid others might not make it through. You argued that instead of shortening the text you would rather give the option to skip certain sections but this is not done. I would also recommend you to move the Methods to the supplementary section but I will leave the decision to you. However, if you decide not to do so you please indicate at the beginning of section 3 that sections 3.1.1-3.1.4, and 3.2-3.4 can be skipped for a less detailed focus on technical aspects.*

We understand the concerns of the Editor and the Reviewer. We were hoping that the sentence "We provide technical descriptions of the steps in the next four subsections but first motivate them here" was describing the intentions of the subsections sufficiently, but we realize that this is not the case. Since the manuscript has become longer during the review process to incorporate the suggestions of the Reviewers, we follow the advise of the Editor in our revised manuscript version and have moved Sect. 3.1.1 – 3.1.4 to the supplemental information. We only retain the most crucial information from these subsections in the revised Sect. 3.1, in particular the structure of the PSM and the properties of the IQD. In the beginning of the revised Sect. 3.1, we refer to the supplemental information for an enhance description of the algorithm.

*6: I believe you should delete the comma after "allows accounting for non-climatic processes,"*

We removed the comma.

*21: Define ka*

We changed the text in brackets to "*LGM, ∼21 ka where ka stands for 'kilo-annum', i.e., thousands of years ago*" in accordance with the submission guidelines.

*27: 100 m is a very low-end estimate, 130 m is more realistic*

We apologize for the typo. We changed it to 130 m.

*33: references should appear in alphabetical order, here and elsewhere*

We have adapted the order of references where needed.

*40: time slices - please mention some examples of time periods.*

We have added "*such as the LGM and the mid-Holocene*".

*70: insert "to" before proxies*

We added "to" before proxies.

*78: again, delete comma after "time series".*

We removed the comma.

*111: replace "than the other" by "than in the other"*

We added "*in*".

*132: delete "do" in "do either"*

We assume that this comment refers to line 123 and we removed "do" in this sentence.

*130: condense to "3K in TRACE-GHG and in FAMOUS"*

We changed the beginning of the sentence to "*With* ~*1 K in TraCE-ORB and* ~*3 K in TRACE-GHG and FAMOUS, ...*".

*141: replace parts by periods*

We adapted the sentence as suggested.

*142-143: same as in line 107*

We were unable to find a comment that this comment could refer back to and, therefore, did not change the text in line 142-143 of the previous tracked changes document.

*163: insert comma after "clustered"*

We added a comma as suggested.

*167-169: As I said above, I agree with reviewer 2 that the methods section is too large. You argue that instead of shortening the text you would rather give the option to skip certain sections but this is not specified. If you decide to go for this option it should be mentioned here.*

As described above, we have moved Sect. 3.1.1 – 3.1.4 to the supplemental information and only retain the most important information from them in the revised Sect. 3.1.

*181: rephrase "We compare measured and forward-modeled proxy time series" to avoid repetition with previous sentence*

We rephrased the sentence to "*We perform this comparison in temperature units ...*".

*184-185: when referring to the patterns here please include "temporal"*

We added "*temporal*" in the lines suggested by the Editor and afterwards in the sentence "*In contrast, temporal pattern components assess ...*".

*194: replace "a simulation could simulate" by "a model could simulate"*

As we assess the deviation between simulations and reconstructions, we have changed the sentence to "*For example, a simulation could reproduce the reconstructed spatio-temporal temperature pattern accurately but receive a poor score due to an under-estimation of the LGM-to-Holocene temperature change.*"

*198-199: Should the description of the Monte Carlo not be included in (1) already (before the decomposition into patterns and magnitudes)?*

We agree with this suggestion by the Editor and have moved the information from the two sentences to (1) in the revised manuscript. To incorporate this change, we have adapted the first two sentences in (3) as follows: "*The decompositions in step 2 result in probability distributions of forward-modeled proxy time series and the corresponding reconstructed SST records because we account for chronological uncertainties and include a noise process in the PSM. We quantify the deviations between these probability distributions with a distance function that takes into account the full probability distributions ...*".

*207: Here I assume lack of understanding in how climatic effects translate into proxies (uncertainty in PSMs) should also be taken into account.*

Indeed, uncertainties in the PSM formulation can also influence the deviations between forward-modeled proxy time series and reconstructions. We have included these uncertainties by rephrasing the two first two sentences in (4) as follows: "*Deviations between forward-modeled proxy time series and reconstructions can depend strongly on the unknown manifestation of non-climatic influences in the measured proxies and uncertainties in the PSM structure. Assuming that most non-climatic processes and PSM uncertainties are uncorrelated between proxy records, the influence of these processes can be reduced by spatially averaging deviations computed for individual proxy records.*"

*216: Why use C and not T to refer to the SST field?*

We changed C to T.

*220-221: How about regions with strong fronts?*

Indeed, the interpolation method can have a non-negligible influence in regions with strong fronts. However, the resolution of most simulations of the deglaciation is coarse (typical grid spacings are between 1° and 5°) such that the representation of small-scale structures in the simulations is limited. Therefore, the representation of regions with strong fronts is likely more affected by the resolution of the employed models than by the interpolation method. We already note in the discussion (Sect. 5.2) that physically-motivated downscaling could help to better represent heterogeneous regions such as coastal upwelling zones. In the revised manuscript, we added the following sentences in the description of the spatial interpolation (note that this subsection has been moved to the supplemental information in the revised manuscript as described in the comments above):

"*Only in areas with strong local heterogeneities such as coastal upwelling zones, the interpolation method could potentially impact the resulting time series. However, the coarse resolution of deglacial simulations likely limits the accurate representation of small-scale structures more than the interpolation method (see also Sect. 5.2 in the main manuscript).*"

*236-240: This explanation on why additive noise is used could be moved to the discussion to shorten this section.*

In the revised manuscript, this explanation has been shifted to the supplemental information, while we keep the discussion of improvements of the PSM structure in the third paragraph of Sect. 5.2.

*274: Equ should be Eq hereafter*

We changed Equ to Eq in accordance with the submission guidelines.

*285: This sentence explaining the IQD name should appear before and not as an isolated paragraph*

We apologize for a rendering issue in the difference document. This sentence was the first sentence of the last paragraph in this section and not an isolated paragraph. Nevertheless, we agree with the suggestion by the Editor and have included this information in a revised version of the first paragraph of this subsection. Note that we have moved this subsection to the supplemental information (see response to comments above).

*345: When you introduce the reference simulation you could also introduce already here the ground truth it represents.*

We have added "*The temperature time series of the reference simulation at each proxy location serves as the ground truth in the PPE.*" as the second sentence in the paragraph to make the connection between reference simulation and ground truth more explicit. We have kept the description of the FPRR and the ground truth deviations in the last two paragraphs of the section because we think the rationale behind these choices becomes more clear after explaining the structure of the PPEs.

*348: "We call it pseudo proxies" should be "We call it a pseudo proxy"*

We changed "pseudo proxies" to "a pseudo proxy".

*352: Can you specify in the text what you mean by simulation characteristics?*

We have added "*such as parameter configurations and the implementation of boundary conditions*".

*374-375: Replace "the absolute value of the IQD" by "its absolute value"*

We have changed the sentence as suggested.

*446: FRPP should be FPRR*

Thanks for pointing out the typo.

*510: Should this reference to Fig 8a not be to Fig 7a?*

The reference should be to Fig. 7a and Fig. 8a because Fig. 7a contains the deviations whereas Fig. 8a shows the absolute magnitudes of the simulates. Accordingly, we have added a reference to Fig. 7a in the revised manuscript.

*A few final more general comments follow. To the extent you think these three issues are important, I would recommend you to include a comment on these, possibly in the abstract already, in order to increase the paper's interest for the broader community.*
*First, the result you point out in lines 539-540 is quite shocking but also revealing. If I understand correctly it implies that over-tuning a model in a given region (for instance via freshwater fluxes) penalises the simulation in others, pointing to the fact that the driving mechanism in the simulation might not be as simple as freshwater fluxes. Lines 696-703 below also relate to this, and I very much agree with this conclusion.*

Indeed, the results hint at limitations from over-tuning a model to proxies from a specific region. As stated by the Editor, we discuss this in Sect. 5.2. To emphasize this result more, we added statements in the abstract ("*The ranking of the simulations differs substantially between the considered regions and timescales, which suggests that optimizing for agreement with the temporal patterns of a small set of proxies might be insufficient for capturing the spatial structure of the deglacial temperature variability.*") and the conclusions ("*This suggests that optimizing for agreement with the temporal patterns of a specific proxy or reconstructions from a small region might be an inadequate strategy for capturing the spatial structure of millennial-scale temperature patterns during the deglaciation.*").

*Second, I might have skipped something but how does internal climate variability affect the method? What happens if millennial-scale variability is entirely internally driven?*

In the case of entirely internally driven millennial-scale variability during the deglaciation, the *millennial pattern* metric does no longer provide meaningful information because it assumes temporal alignment of reconstructions and simulations. This cannot be expected for internally-driven fluctuations. On the other hand, the *millennial magnitude* metric still provides useful information for evaluating the strength of millennial-scale fluctuations. We believe that Sect. 5.2 is the best place to discuss this question and that it is not important enough to highlight it in the abstract or conclusions. While the second-to-last paragraph in Sect. 5.2 already included a discussion of this issue, we rephrased it to emphasize the question of internally driven variability stronger:

"*As the PPEs and the real-world application have shown, the pattern IQDs are sensitive to the timing of timescale-dependent temperature fluctuations. Therefore, they are only meaningful if the goal of a simulation is to reproduce a specific succession of variations observed in reconstructions. Temporal alignment cannot be expected for internally driven variations such as spontaneous millennial-scale fluctuations (…), and in the presence of boundary conditions with large spatio-temporal uncertainties like deglacial meltwater fluxes. In these cases, the magnitude IQDs, which are insensitive to the timing of fluctuations, could be combined with a more insightful measure for temporal patterns, e.g., based on the similarity of spatial relationships in reconstructed and forward-modeled proxy time series (…).*"

*Finally, how specific is the method to the last deglaciation? To what extent would it be applicable for example to other periods (give examples)?*

The last paragraph of Sect. 5.2 states that the method can be applied to other continuous variables (e.g., oxygen isotopes) and other periods (e.g., penultimate deglaciation, the glacial inception, or the last glacial cycle). To emphasize this point, we have included additional sentences in the abstract ("*Additionally, the algorithm can be applied to variables like oxygen isotopes and climate transitions such as the penultimate deglaciation and the last glacial inception.*") and the conclusions ("*In addition to assessing the temperature evolution during the last deglaciation, the proposed method can be applied to other continuous variables, e.g., oxygen isotopes, and other periods with climate transitions such as the penultimate deglaciation and the last glacial inception.*") to state these possible applications of our proposed method.

*Notification to the authors (from file validation):*
*With the next file upload request, please update the "Competing interests" as follows: At least one of the (co-)authors is a member of the editorial board of Climate of the Past*

We apologize for not including this statement before and have added it in the revised manuscript.